# The Double Dilemma in Multi-Task Radiology Report Generation: A Gradient Dynamics Analysis and Solution

Erjian Zhang [1]   Yatong Hao [1]   Liejun Wang [1 2]   Zhiqing Guo [1 2]

## Abstract

While multi-task learning based automatic radiology report generation (RRG) is widely adopted to ensure clinical consistency, most focus on architectural designs yet remain limited to coarse linear scalarization strategies. These strategies cannot effectively balance the hard constraints of discriminative clinical supervision with the smoothness requirements of report generation. To address these problems, we analyze the failure mechanism of linear scalarization from the perspective of gradient dynamics, utilizing the stochastic differential equation (SDE) framework to characterize it as a "Double Dilemma" of drift term deviation and diffusion term decay. Based on this, we propose a backbone-agnostic optimizer named **C**onflict-**A**verse **M**agnitude-**E**nhanced **Grad**ient Descent (CAME-Grad). Through conflict-averse direction rectification and magnitude-enhanced energy injection, the algorithm not only ensures geometric validity, but also avoids local optimal solutions. Then, the adaptive gradient fusion mechanism is used to establish a dynamic balance between the theoretical optimal direction and the task-specific inductive bias. Experiments show that as a universal plug-and-play optimizer, CAME-Grad brings substantial and consistent improvements across eight diverse RRG methods, elevating overall clinical efficacy performance by an average of 2.3% on MIMIC-CXR and 1.9% on IU X-Ray. Our code is available at https://github.com/vpsg-research/CAME-Grad.

[1]School of Computer Science and Technology, Xinjiang University, Urumqi, China [2]Xinjiang Multimodal Intelligent Processing and Information Security Engineering Technology Research Center, Urumqi, China. Correspondence to: Liejun Wang <wljxju@xju.edu.cn>, Zhiqing Guo <guozhiqing@xju.edu.cn>.

*Proceedings of the 43rd International Conference on Machine Learning*, Seoul, South Korea. PMLR 306, 2026. Copyright 2026 by the author(s).

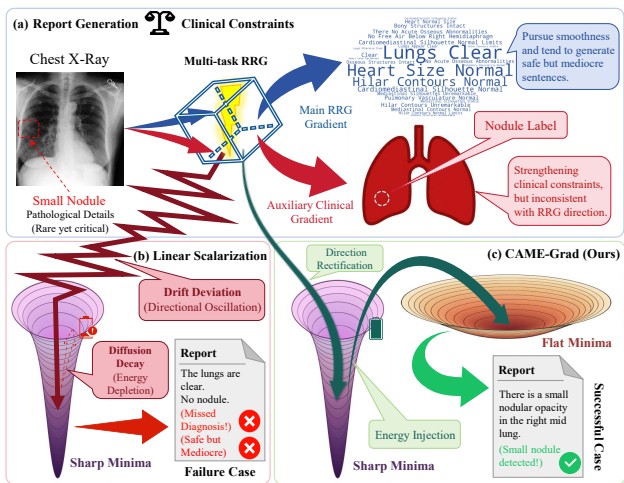

*Figure 1.* The "Double Dilemma" in RRG multi-task optimization and its resolution via CAME-Grad. **(a)** In multi-task RRG, there is an intrinsic conflict between report generation and clinical constraints. **(b)** Under linear scalarization, this conflict simultaneously induces drift term deviation and diffusion term decay. **(c)** CAME-Grad employs direction rectification to ensure geometric validity and energy injection to escape sharp minima.

## 1. Introduction

Automated radiology report generation (RRG) aims to reduce the heavy workload of radiologists and improve diagnostic efficiency. Early methods (Vinyals et al., 2015; Chen et al., 2020) predominantly adopted a single-task learning paradigm, relying solely on a single text likelihood supervision to optimize generation quality. However, this single supervision signal often fails to guarantee clinical diagnostic accuracy. To address this, existing works have widely shifted to multi-task learning (MTL) paradigms to enhance feature representation and clinical consistency.

The technical evolution of this paradigm introduces diverse clinical constraint tasks, encompassing disease classification strategies aiming to explicitly enhance diagnostic accuracy (Li et al., 2025), image-text alignment mechanisms designed to bridge cross-modal disparities (Wang et al., 2024), and retrieval enhancement modules assisting generation by incorporating external medical knowledge (Song et al., 2025). Although these sophisticated architectural designs improve

performance, existing works largely overlook the underlying gradient dynamics of multi-task optimization, defaulting to simple linear scalarization strategies. This coarse optimization strategy fails to effectively reconcile the intrinsic conflict between two types of tasks in RRG. In particular, strengthening the hard constraints of discriminative clinical supervision often leads to the degradation of generated report quality (Jin et al., 2024), while prioritizing the smoothness of language models makes it difficult to capture rare and critical pathological details (Li et al., 2025). Consequently, this dilemma limits the full realization of model potential.

To bridge this gap, it is imperative to deeply analyze the geometric roots underlying the suboptimality of existing RRG multi-task optimization. From the perspective of gradient dynamics, the optimization dynamics in RRG inherently involve the interplay between the drift term driven by the first-order moment and the diffusion term arising from the second-order covariance. Motivated by this, we leverage the stochastic differential equation (SDE) framework (Mandt et al., 2017) to characterize the failure mechanism of linear scalarization in handling these dynamics as a "Double Dilemma" (Figure 1). Specifically, under traditional linear scalarization strategies, the intrinsic conflict between the RRG primary report generation task and auxiliary clinical constraint tasks causes the resultant force direction to deviate from the Pareto optimal trajectory, thereby inducing drift term deviation. This fundamentally constitutes directional instability in optimization dynamics, manifesting as severe directional oscillation that hinders optimization efficiency. Simultaneously, this conflict leads to a structural collapse in gradient magnitude, thereby triggering diffusion term decay. This fundamentally constitutes a deficiency in exploration kinetic energy, manifesting as severe energy depletion, which renders the model unable to escape local optimal solutions.

Based on this theoretical insight, we propose **C**onflict-**A**verse **M**agnitude-**E**nhanced **Grad**ient Descent (CAME-Grad). As a backbone-agnostic optimizer, CAME-Grad directly replaces traditional linear scalarization strategies without modifying the core network architecture, aiming to resolve this "Double Dilemma" by reshaping the optimization dynamics via three cascaded stages. First, we employ Conflict-Averse Direction Rectification to suppress destructive interference, thereby mitigating the drift term deviation and establishing geometric validity in the tangent space. Second, we introduce a Magnitude-Enhanced Energy Injection mechanism to actively restore and enhance the gradient magnitude. This step compensates for the diffusion term decay by injecting escape kinetic energy to drive the model from sharp minima toward flat minima. Finally, we implement Adaptive Gradient Fusion to balance the theoretical optimal direction with task-specific inductive

biases, effectively preventing the catastrophic forgetting of semantic features. The main contributions of this paper are summarized as follows:

- We investigate the failure mechanism of linear scalarization from the perspective of gradient dynamics, utilizing the SDE framework to characterize it as a "Double Dilemma" of drift term deviation and diffusion term decay, thereby revealing the geometric roots of suboptimality in RRG multi-task optimization.

- We propose CAME-Grad, a gradient optimization algorithm designed for multi-task RRG. Formulated as a backbone-agnostic optimizer, it enables plug-and-play integration into diverse model architectures.

- We conduct extensive evaluations of CAME-Grad on the MIMIC-CXR and IU X-Ray datasets across eight representative RRG methods, demonstrating average clinical efficacy improvements of 2.3% and 1.9%, respectively.

## 2. Related Work

### 2.1. Radiology Report Generation

Radiology report generation (RRG) aims to automatically generate accurate and coherent diagnostic reports from medical images. Unlike general natural image captioning tasks (Vinyals et al., 2015), RRG requires processing longer text sequences and capturing subtle pathological details. Early research (Jing et al., 2018; Li et al., 2019; Chen et al., 2020; 2021) primarily adopted single-task learning paradigms, utilizing mechanisms such as co-attention, prior knowledge injection, or shared memory to bridge the visual-semantic gap. However, relying solely on a single generation supervision signal, models struggle to align these complex cross-modal representations and are prone to generating hallucinations for long-tail disease descriptions.

To overcome single-task bottlenecks, multi-task learning (MTL) has become the mainstream research paradigm. Specifically, disease classification tasks (Jin et al., 2024; Li et al., 2025) are introduced to inject clinical priors, while image-text alignment methods (Wang et al., 2022; 2024; Yan et al., 2021; Li et al., 2023) strive to maintain anatomical consistency and learn discriminative features. Additionally, retrieval-based enhancement methods (Zhou et al., 2025; Song et al., 2025) leverage external knowledge bases or patient historical data to supplement diagnostic information. Despite the increasingly sophisticated architectures, existing works still predominantly adopt static linear scalarization in their optimization strategies. This overlooks the complex gradient dynamics among multiple tasks, thereby limiting the release of the potential inherent in RRG models.

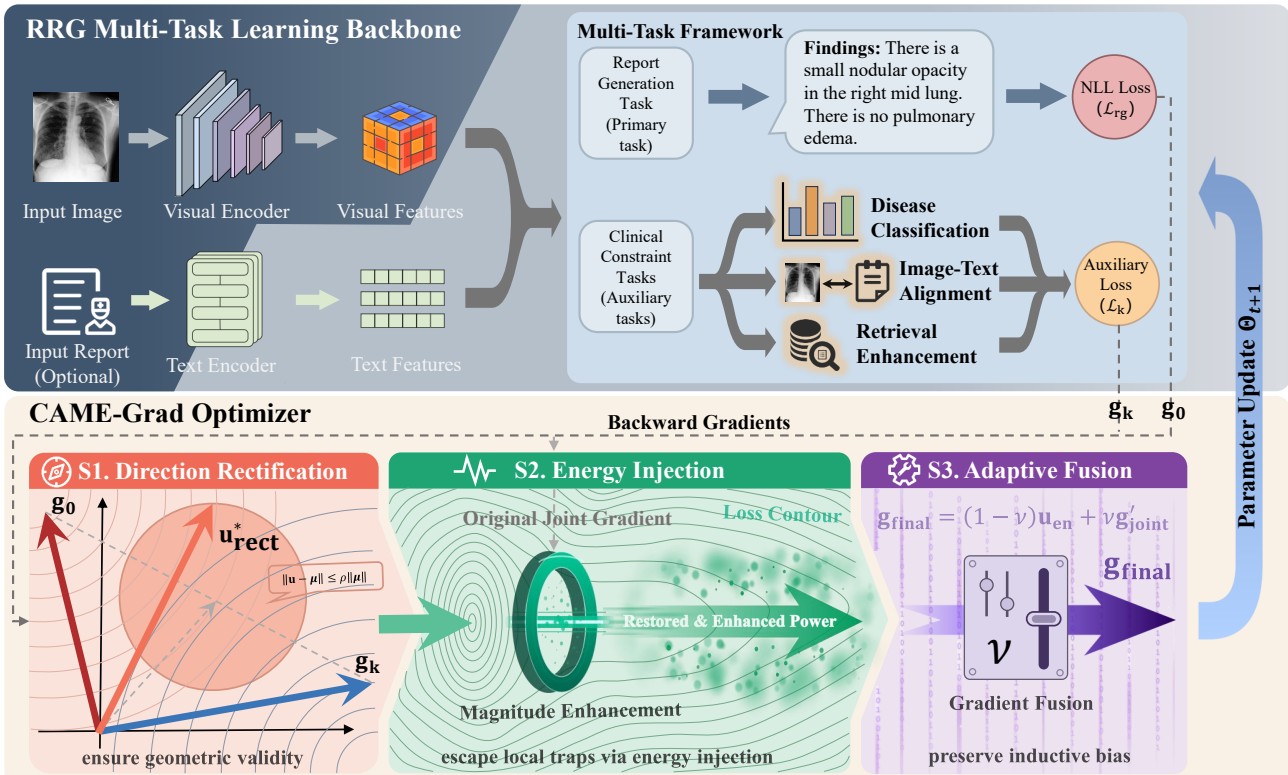

*Figure 2.* Architecture of the proposed CAME-Grad optimizer. (Top) The multi-task backbone integrates disease classification, image-text alignment, and retrieval enhancement as clinical constraints. (Bottom) The CAME-Grad optimizer operates via three stages. (S1) Direction rectification resolves drift deviation via geometric projection. (S2) Energy injection reverses diffusion decay by restoring and enhancing gradient magnitude. (S3) Adaptive fusion balances the theoretically optimal direction with task-specific inductive biases.

## 2.2. Gradient Dynamics and Optimization Dilemmas

To mitigate gradient conflicts by rectifying update directions, algorithms evolved from classic projection (Sener & Koltun, 2018; Yu et al., 2020), Pareto optimization (Liu et al., 2021), and game-theoretic perspectives (Navon et al., 2022) to recent fast adaptive approaches (Liu et al., 2023), task priority quantification (Jeong & Yoon, 2024), and bi-level consistency (Qin et al., 2025). Meanwhile, magnitude regulation strategies advanced from normalization (Chen et al., 2018) and uncertainty weighting (Kendall et al., 2018) to selective grouping (Wei & Hu, 2024; Jeong & Yoon, 2025) and prioritized multipliers (Cheng et al., 2025), enhancing the implicit regularization of stochastic gradient descent (SGD). Nevertheless, directly applying these paradigms to RRG encounters dynamical conflicts where direction rectification sacrifices diffusion kinetic energy, while strategies solely boosting magnitude struggle to ensure geometric validity in direction. To address this dilemma, CAME-Grad leverages the SDE framework to reshape the dynamics of the drift term and the diffusion term. By establishing a novel evolution mechanism, it enhances the diffusion kinetic energy within the geometrically valid tangent space of the manifold, thereby fully releasing the potential in RRG models.

## 3. Method

In this section, we first provide a theoretical analysis of gradient dynamics in multi-task RRG. Addressing the "Double Dilemma" characterized by the SDE formulation, we propose the CAME-Grad gradient optimization algorithm. The overall architecture is illustrated in Figure 2.

### 3.1. Multi-Task RRG

Let $\mathcal{D} = \{(I_n, Y_n)\}_{n=1}^N$ denote a dataset with $N$ samples, where $I_n$ represents a medical image, and $Y_n = \{y_1, \ldots, y_T\}$ denotes the corresponding report sequence.

We utilize a visual encoder $\mathcal{E}_\phi$ and a text decoder $\mathcal{D}_\psi$ to construct the primary report generation task. Let $\Theta_{rg} = \{\phi, \psi\}$ denote the parameters for this task. The encoder maps $I_n$ to latent features $\mathbf{Z}_n = \mathcal{E}_\phi(I_n)$, and the decoder models the conditional probability $P_{\Theta_{rg}}(Y_n \mid I_n)$. The primary objective is to minimize the negative log-likelihood:

$$\mathcal{L}_{rg}(\Theta_{rg}) = -\frac{1}{N}\sum_{n=1}^N \log P_{\Theta_{rg}}(Y_n \mid I_n). \quad (1)$$

We introduce $K$ auxiliary clinical constraint tasks. These

tasks map the pooled visual features $\bar{\mathbf{z}}_n = \text{Pool}(\mathbf{Z}_n)$ to discrete clinical labels $l_n \in \{0,1\}^C$ via task-specific projection heads $f_{\theta_k}$. The optimization objective for these constraints is formulated as:

$$\mathcal{L}_k(\phi, \theta_k) = -\frac{1}{N} \sum_{n=1}^{N} \sum_{c=1}^{C} \Big[ l_{n,c} \log \sigma(f_{\theta_k}(\bar{\mathbf{z}}_n)_c)$$
$$+ (1 - l_{n,c}) \log(1 - \sigma(f_{\theta_k}(\bar{\mathbf{z}}_n)_c)) \Big]. \qquad (2)$$

To establish a unified optimization framework, we denote the total parameter set as $\Theta = \Theta_{rg} \cup \{\theta_1, \dots, \theta_K\}$. We index the primary task as 0 (i.e., $\mathcal{L}_0 \triangleq \mathcal{L}_{rg}$), and auxiliary tasks as $1, \dots, K$. The overall multi-task objective is a linear weighted combination of all task losses:

$$\mathcal{L}_{joint}(\Theta) = \sum_{i=0}^{K} \omega_i \mathcal{L}_i(\Theta), \qquad (3)$$

where $\omega_i$ represents predefined static task weights.

### 3.2. Theoretical Analysis of Gradient Dynamics

Let $\mathbf{g}_i = \nabla_\Theta \mathcal{L}_i(\Theta) \in \mathbb{R}^d$ be the gradient vector for the $i$-th task regarding shared parameters $\Theta$. In standard SGD using the linear scalarization strategy (Sener & Koltun, 2018), the joint update direction $\mathbf{g}_{joint}$ comes from the linear superposition of task gradients:

$$\mathbf{g}_{joint} = \sum_{i=0}^{K} \omega_i \mathbf{g}_i, \qquad (4)$$

where $\omega_i$ are static scalar weights. In this section, we analyze the theoretical flaws of this static linear scalarization strategy in handling the intrinsic conflict of RRG multi-task learning, from both geometric and dynamical dimensions.

#### 3.2.1. GEOMETRIC QUANTIFICATION OF CONFLICT

We define the gradient conflict between the primary task (indexed as 0) and an auxiliary task $k \in \{1, \dots, K\}$ as the occurrence of a negative cosine similarity between their respective gradients $\mathbf{g}_0$ and $\mathbf{g}_k$. To analyze the energy interaction, we expand the squared norm of the linear combination:

$$\|\omega_0 \mathbf{g}_0 + \omega_k \mathbf{g}_k\|^2$$
$$= \omega_0^2 \|\mathbf{g}_0\|^2 + \omega_k^2 \|\mathbf{g}_k\|^2 + 2 \underbrace{\omega_0 \omega_k (\mathbf{g}_0^\top \mathbf{g}_k)}_{\text{interaction term } \mathcal{I}_k}. \qquad (5)$$

The RRG primary report generation task requires a smooth semantic manifold, whereas auxiliary clinical constraint tasks induce rigid, discretized feature structures. This intrinsic conflict causes persistent destructive interference, manifested as a significantly negative interaction term $\mathcal{I}_k \ll 0$, as illustrated in Figure 3 and generalized in Appendix A.

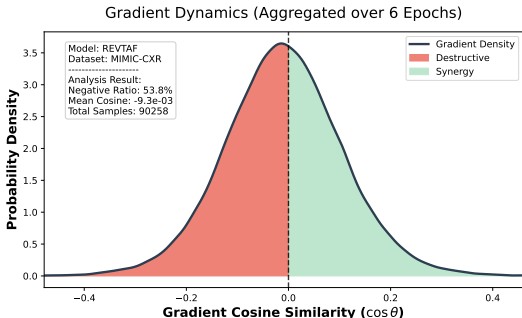

*Figure 3.* The substantial negative ratio of 53.8% quantitatively confirms the intrinsic conflict between the report generation and clinical constraint tasks.

#### 3.2.2. CHARACTERIZING THE DOUBLE DILEMMA VIA SDE

To reveal the dynamical consequences of this intrinsic conflict, we model the optimization dynamics of SGD using SDE. The discrete SGD update rule $\Theta_{t+1} = \Theta_t - \eta\hat{\mathbf{g}}(\Theta_t)$ approximates the following continuous-time SDE process:

$$d\Theta_t = \underbrace{-\mathbf{g}_{joint}(\Theta_t)dt}_{\text{drift term}} + \underbrace{\sqrt{\eta\Sigma(\Theta_t)}d\mathbf{W}_t}_{\text{diffusion term}}, \qquad (6)$$

where $\mathbf{g}_{joint}$ is the expected joint gradient, $\Sigma$ is the covariance matrix of gradient noise, and $\mathbf{W}_t$ represents Brownian motion. In this framework, the drift term controls the convergence direction, while the diffusion term provides exploration noise to escape local minima.

Under the linear combination $\mathbf{g}_{joint} = \sum_{i=0}^{K} \omega_i \mathbf{g}_i$, the energy of the joint gradient is bound by the interaction term from Section 3.2.1:

$$\|\mathbf{g}_{joint}\|^2 \approx \omega_0^2 \|\mathbf{g}_0\|^2 + \sum_{k=1}^{K} \omega_k^2 \|\mathbf{g}_k\|^2 + 2 \sum_{k=1}^{K} \mathcal{I}_k, \qquad (7)$$

where inter-auxiliary interference is neglected as secondary. Since linear weights are strictly positive, linear scalarization fails to mathematically decouple destructive gradient interference. This limitation characterizes a "Double Dilemma" within our SDE-based mechanistic hypothesis. Severe gradient conflicts cause the resultant force $\mathbf{g}_{joint}$ to diverge from the Pareto optimal direction, inducing drift deviation. Simultaneously, the magnitude collapse caused by negative interactions directly results in diffusion decay, stifling exploration noise.

### 3.3. CAME-Grad Gradient Optimization Algorithm

As a backbone-agnostic optimizer, CAME-Grad serves as a direct substitute for traditional linear scalarization strategies, effectively resolving this "Double Dilemma" by reshaping the optimization dynamics via three integrated stages, as summarized in Algorithm 1.

---

**Algorithm 1** CAME-Grad Optimization Process

---

**Input:** Dataset $\mathcal{D}$, Parameters $\Theta$, Learning rate $\eta$
**Hyperparams:** Task weights $\omega$, Radius $\rho$, Gain factor $\kappa$, Fusion coefficient $\nu$
**while** not converged **do**
    Sample batch $\mathcal{B} \sim \mathcal{D}$; Compute task gradients $\mathbf{g}_i \leftarrow \nabla_\Theta \mathcal{L}_i(\Theta)$
    Compute joint gradient $\mathbf{g}_{joint} \leftarrow \sum_{i=0}^K \omega_i \mathbf{g}_i$ and mean gradient $\boldsymbol{\mu} \leftarrow \frac{1}{K+1} \sum_{i=0}^K \mathbf{g}_i$
    *// Stage 1: Conflict-Averse Direction Rectification*
    Solve dual problem for $\boldsymbol{\alpha}^*$ and compute rectified direction $\mathbf{u}_{rect}^*$ (closed-form)
    *// Stage 2: Magnitude-Enhanced Energy Injection*
    Calculate target magnitude: $\tau_{mag} \leftarrow \kappa \|\mathbf{g}_{joint}\|$
    Compute enhanced gradient: $\mathbf{u}_{en} \leftarrow \mathbf{u}_{rect}^* \cdot \frac{\tau_{mag}}{\|\mathbf{u}_{rect}^*\| + \epsilon}$
    *// Stage 3: Adaptive Gradient Fusion*
    Compute final direction: $\mathbf{g}_{final} \leftarrow (1 - \nu)\mathbf{u}_{en} + \nu(\kappa \mathbf{g}_{joint})$
    Update parameters: $\Theta \leftarrow \Theta - \eta \mathbf{g}_{final}$
**end while**

---

**Stage 1: Conflict-Averse Direction Rectification** First, to ensure geometric validity across all tasks, we search for a vector $\mathbf{u}$ that maximizes the worst-case local improvement. Inspired by the optimization principles of CAGrad (Liu et al., 2021), this search happens within a trust region centered at the mean gradient $\boldsymbol{\mu} = \frac{1}{K+1} \sum_{i=0}^K \mathbf{g}_i$. We define the primal problem as:

$$\max_{\mathbf{u} \in \mathbb{R}^d} \min_{i \in \{0, \ldots, K\}} \mathbf{g}_i^\top \mathbf{u} \quad \text{s.t.} \quad \|\mathbf{u} - \boldsymbol{\mu}\| \leq \rho \|\boldsymbol{\mu}\|, \quad (8)$$

where $\rho \in [0, 1)$ is a hyperparameter controlling the radius. To guarantee global convergence stability, our rectification step strictly adheres to the trust region assumption, ensuring that the corrected gradient $\mathbf{u}$ resides within a bounded proximity of the mean gradient $\boldsymbol{\mu}$. This constraint ensures optimization stability by limiting the deviation from the mean gradient. Instead of resolving the intractable high-dimensional conflicts directly in the parameter space, we introduce dual variables $\boldsymbol{\alpha} \in \Delta^{K+1}$ and a weighted gradient $\mathbf{g}_{\boldsymbol{\alpha}} = \sum_{i=0}^K \alpha_i \mathbf{g}_i$ to transform this primal problem into a solvable convex optimization problem over the unit simplex, where the dual objective is formulated as:

$$\min_{\boldsymbol{\alpha} \in \Delta^{K+1}} \mathcal{F}(\boldsymbol{\alpha}) := \mathbf{g}_{\boldsymbol{\alpha}}^\top \boldsymbol{\mu} + \sqrt{\xi} \|\mathbf{g}_{\boldsymbol{\alpha}}\|, \quad (9)$$

where $\xi = \rho^2 \|\boldsymbol{\mu}\|^2$. Furthermore, by solving this tensorized dual problem directly on the GPU, CAME-Grad inherently avoids the host-device communication bottlenecks typical of $\mathcal{O}(d)$ gradient manipulations, ensuring a negligible computational overhead during training to obtain the optimal weights $\boldsymbol{\alpha}^*$. Subsequently, we recover the optimal rectified

direction $\mathbf{u}_{rect}^*$ via its closed-form solution:

$$\mathbf{u}_{rect}^* = \boldsymbol{\mu} + \frac{\sqrt{\xi}}{\|\mathbf{g}_{\boldsymbol{\alpha}^*}\|} \mathbf{g}_{\boldsymbol{\alpha}^*}. \quad (10)$$

This direction rectification process is intended to reshape the drift coefficient $-\mathbf{g}_{joint}$ in the SDE, aiming to keep the update trajectory within the geometrically valid tangent space of the manifold, thereby mitigating the drift deviation caused by task conflicts.

**Stage 2: Magnitude-Enhanced Energy Injection** While $\mathbf{u}_{rect}^*$ mitigates directional conflicts, the constrained optimization significantly compresses its magnitude. To counteract this magnitude collapse, we introduce a magnitude enhancement mechanism to define a target magnitude $\tau_{mag}$ through a dual-level process. First, we restore the magnitude to the baseline level of the original joint gradient $\|\mathbf{g}_{joint}\|$ to retrieve the lost fundamental diffusion kinetic energy. Second, upon this basis, we introduce a gain factor $\kappa \geq 1$ for further enhancement to inject additional exploration noise. The target magnitude is calculated as:

$$\tau_{mag} = \kappa \cdot \|\mathbf{g}_{joint}\|. \quad (11)$$

Subsequently, we restore and enhance the magnitude of the rectified direction $\mathbf{u}_{rect}^*$ to the target level $\tau_{mag}$, yielding the enhanced gradient $\mathbf{u}_{en}$, formulated as:

$$\mathbf{u}_{en} = \mathbf{u}_{rect}^* \cdot \frac{\tau_{mag}}{\|\mathbf{u}_{rect}^*\| + \epsilon}, \quad (12)$$

where $\epsilon$ is a small constant for numerical stability. By strictly enforcing the target magnitude $\tau_{mag}$, CAME-Grad is designed to compensate for the diffusion intensity governed by $\Sigma(\Theta_t)$ in the SDE. This step seeks to counteract the diffusion decay induced by energy depletion, facilitating the exploration needed to escape sharp minima toward flatter loss basins.

**Stage 3: Adaptive Gradient Fusion** Pure mathematical rectification might overly orthogonalize the gradients. This causes the loss of weak but important feature signals, such as gradients for long-tail tokens. To keep these task-specific inductive biases, we design an adaptive fusion mechanism. We use a fusion coefficient $\nu \in [0, 1]$ to linearly interpolate between the theoretical optimal gradient $\mathbf{u}_{en}$ and the magnitude-enhanced original joint gradient $\mathbf{g}_{joint}' = \kappa \mathbf{g}_{joint}$:

$$\mathbf{g}_{final} = (1 - \nu)\mathbf{u}_{en} + \nu \mathbf{g}_{joint}'. \quad (13)$$

The final parameter update rule with learning rate $\eta$ is:

$$\Theta_{t+1} = \Theta_t - \eta \cdot \mathbf{g}_{final}. \quad (14)$$

This mechanism strikes a balance between the theoretically optimal direction and task-specific inductive biases. A small

*Table 1.* Performance comparison on the MIMIC-CXR test set. Best results for each baseline pair are highlighted in bold and ↑ indicates higher is better. All models are reproduced following the unified experimental setup described in Section 4.3. CE Average denotes the arithmetic mean of Precision, Recall, and F1-score.

| MODEL | PUBLICATION | CE METRICS | | | NLG METRICS | | | | | | | CE AVG. ↑ |
|---|---|---|---|---|---|---|---|---|---|---|---|---|
| | | PREC. ↑ | REC. ↑ | F1 ↑ | B-1 ↑ | B-2 ↑ | B-3 ↑ | B-4 ↑ | MTR ↑ | R-L ↑ | | |
| WCL | EMNLP'21 | 0.382 | 0.298 | 0.312 | 0.329 | 0.203 | 0.136 | 0.098 | 0.135 | 0.277 | | 0.331 |
| + CAME-GRAD | | **0.424** | **0.324** | **0.344** | **0.349** | **0.219** | **0.150** | **0.108** | **0.142** | **0.284** | | **0.364** |
| XPRONET | ECCV'22 | 0.382 | 0.304 | 0.314 | 0.325 | 0.202 | 0.137 | **0.099** | 0.134 | 0.260 | | 0.333 |
| + CAME-GRAD | | **0.391** | **0.309** | **0.322** | **0.329** | **0.204** | **0.138** | 0.099 | **0.135** | **0.261** | | **0.341** |
| DCL | CVPR'23 | 0.265 | 0.246 | 0.236 | 0.234 | 0.136 | 0.087 | 0.061 | 0.108 | 0.211 | | 0.249 |
| + CAME-GRAD | | **0.319** | **0.298** | **0.286** | **0.280** | **0.162** | **0.104** | **0.074** | **0.122** | **0.220** | | **0.301** |
| PROMPTMRG | AAAI'24 | 0.505 | 0.518 | 0.482 | **0.396** | **0.236** | 0.152 | **0.105** | 0.155 | 0.265 | | 0.502 |
| + CAME-GRAD | | **0.514** | **0.530** | **0.491** | 0.396 | 0.236 | **0.153** | 0.105 | **0.157** | **0.266** | | **0.512** |
| CAMANET | JBHI'24 | 0.405 | 0.303 | 0.321 | **0.327** | 0.202 | 0.136 | 0.097 | 0.134 | 0.274 | | 0.343 |
| + CAME-GRAD | | **0.410** | **0.313** | **0.330** | 0.327 | **0.203** | **0.137** | **0.099** | **0.135** | **0.275** | | **0.351** |
| DDATR | TMI'25 | 0.408 | 0.444 | 0.418 | 0.393 | 0.235 | 0.152 | 0.106 | 0.158 | 0.267 | | 0.423 |
| + CAME-GRAD | | **0.432** | **0.463** | **0.438** | **0.405** | **0.245** | **0.161** | **0.113** | **0.161** | **0.273** | | **0.444** |
| TGRG | MEDIA'25 | 0.457 | 0.435 | 0.417 | 0.423 | 0.258 | 0.164 | 0.109 | 0.162 | 0.284 | | 0.436 |
| + CAME-GRAD | | **0.493** | **0.466** | **0.450** | **0.438** | **0.268** | **0.173** | **0.117** | **0.169** | **0.285** | | **0.470** |
| REVTAF | ICCV'25 | 0.615 | 0.617 | 0.589 | 0.465 | 0.319 | 0.236 | 0.184 | 0.199 | **0.336** | | 0.607 |
| + CAME-GRAD | | **0.633** | **0.637** | **0.607** | **0.466** | **0.320** | **0.237** | **0.185** | **0.200** | 0.336 | | **0.626** |

$\nu$ (e.g., 0.2) effectively avoids conflicts and ensures sufficient magnitude while preserving fine-grained semantic information, thereby fully releasing the potential inherent in RRG models.

Comprehensive theoretical validations and a terminology glossary are provided in Appendices B and C.

## 4. Experiments

### 4.1. Datasets

MIMIC-CXR (Johnson et al., 2019) serves as the largest public dataset consisting of chest X-ray images and paired radiology reports. To ensure a fair comparison with baseline works, we strictly follow the official split and preprocessing pipeline proposed by Chen et al. (2020). Consequently, the processed dataset contains 270,790 samples for training, 2,130 for validation, and 3,858 for testing. The IU X-Ray dataset (Demner-Fushman et al., 2016), provided by Indiana University, comprises 7,470 frontal and lateral chest X-ray images associated with 3,955 diagnostic reports.

### 4.2. Evaluation Metrics

To comprehensively evaluate the performance of our model, we employ both Natural Language Generation (NLG) and Clinical Efficacy (CE) metrics. For NLG metrics, we report standard natural language processing scores including BLEU-1 to BLEU-4 (B-1 to B-4) (Papineni et al., 2002), METEOR (MTR) (Denkowski & Lavie, 2011), and ROUGE-L (R-L) (Lin, 2004), which primarily measure the

lexical overlap and linguistic fluency at the n-gram level. Regarding CE metrics, following the recommendations of Nicolson et al. (2023), we assess diagnostic accuracy using CheXbert (Smit et al., 2020). This BERT-based labeler extracts 14 distinct observation categories from both generated and reference reports to calculate Precision (Prec.), Recall (Rec.), and F1-score (F1).

### 4.3. Implementation Details

We implement CAME-Grad in PyTorch (Paszke et al., 2019) using a single NVIDIA A40 GPU. To ensure a fair and consistent comparison, we strictly adhere to the official implementations and default configurations reported in the original papers for all baseline models, while introducing two necessary standardizations to align the evaluation protocol. First, regarding model selection, we unify the criterion for saving the best model by utilizing the clinical F1-score on the validation set to prioritize clinical diagnostic accuracy. Second, we standardize the calculation of clinical efficacy metrics across all baselines to the protocol described in Section 4.2, specifically calculating the instance-level F1-score over 14 diseases using CheXbert. Regarding the IU X-Ray dataset, we adopt an adaptive strategy to align with baseline protocols by applying zero-shot inference for cross-domain methods (Jin et al., 2024; Song et al., 2025; Zhou et al., 2025) and conducting standard supervised training on official splits for the remaining baselines (Yan et al., 2021; Wang et al., 2022; Li et al., 2023; Wang et al., 2024; Li et al., 2025). We initialize CAME-Grad with $\rho = 0.5, \kappa = 1.5, \nu = 0.2$ and tailor these hyperparameters

*Table 2.* Performance comparison on the IU X-Ray test set. Best results for each baseline pair are highlighted in bold and ↑ indicates higher is better. All models are reproduced following the unified experimental setup in Section 4.3 where ∗ and † represent zero-shot inference and supervised training respectively. CE Average denotes the arithmetic mean of Precision, Recall, and F1-score.

| MODEL | PUBLICATION | CE METRICS | | | NLG METRICS | | | | | | | CE AVG. ↑ |
|---|---|---|---|---|---|---|---|---|---|---|---|---|
| | | PREC. ↑ | REC. ↑ | F1 ↑ | B-1 ↑ | B-2 ↑ | B-3 ↑ | B-4 ↑ | MTR ↑ | R-L ↑ | | |
| WCL† | EMNLP'21 | 0.490 | 0.498 | 0.491 | 0.325 | 0.195 | 0.137 | 0.101 | 0.137 | 0.318 | | 0.493 |
| + CAME-GRAD | | **0.522** | **0.514** | **0.516** | **0.362** | **0.226** | **0.163** | **0.124** | **0.156** | **0.332** | | **0.517** |
| XPRONET† | ECCV'22 | 0.598 | 0.588 | 0.590 | 0.440 | 0.271 | 0.183 | 0.132 | **0.182** | **0.336** | | 0.592 |
| + CAME-GRAD | | **0.619** | **0.603** | **0.606** | **0.463** | **0.292** | **0.204** | **0.151** | 0.181 | 0.333 | | **0.609** |
| DCL† | CVPR'23 | 0.527 | 0.525 | 0.525 | 0.393 | 0.255 | 0.186 | 0.144 | 0.185 | 0.298 | | 0.526 |
| + CAME-GRAD | | **0.575** | **0.569** | **0.571** | **0.399** | **0.262** | **0.193** | **0.151** | **0.191** | **0.312** | | **0.572** |
| PROMPTMRG∗ | AAAI'24 | 0.200 | 0.210 | 0.197 | **0.419** | **0.248** | 0.158 | 0.106 | **0.161** | **0.314** | | 0.202 |
| + CAME-GRAD | | **0.207** | **0.217** | **0.203** | **0.419** | **0.248** | **0.159** | **0.107** | 0.158 | 0.308 | | **0.209** |
| CAMANET† | JBHI'24 | 0.503 | 0.490 | 0.494 | 0.398 | 0.256 | 0.185 | 0.142 | 0.166 | 0.339 | | 0.496 |
| + CAME-GRAD | | **0.520** | **0.514** | **0.516** | **0.409** | **0.261** | **0.190** | **0.145** | **0.170** | **0.349** | | **0.517** |
| DDATR∗ | TMI'25 | **0.308** | 0.282 | 0.257 | 0.428 | 0.253 | 0.160 | 0.107 | 0.161 | **0.314** | | 0.282 |
| + CAME-GRAD | | 0.281 | **0.308** | **0.259** | **0.433** | **0.257** | **0.164** | **0.110** | **0.163** | 0.313 | | **0.283** |
| TGRG† | MEDIA'25 | 0.508 | 0.497 | 0.500 | 0.471 | 0.302 | 0.204 | **0.146** | 0.189 | **0.392** | | 0.502 |
| + CAME-GRAD | | **0.527** | **0.530** | **0.525** | **0.498** | **0.308** | **0.207** | 0.145 | **0.194** | 0.382 | | **0.527** |
| REVTAF∗ | ICCV'25 | 0.292 | 0.291 | 0.283 | 0.420 | 0.249 | **0.159** | **0.106** | **0.178** | **0.311** | | 0.289 |
| + CAME-GRAD | | **0.304** | **0.302** | **0.294** | **0.424** | **0.251** | **0.159** | 0.105 | 0.177 | 0.310 | | **0.300** |

for each backbone, with detailed configurations provided in Appendices D and E.

## 4.4. Results

### 4.4.1. COMPARISON WITH RRG BASELINES

To verify the effectiveness of CAME-Grad, we conduct experiments on two widely used datasets, namely MIMIC-CXR and IU X-Ray. We compare our method with eight state-of-the-art baselines, including WCL (Yan et al., 2021), XProNet (Wang et al., 2022), DCL (Li et al., 2023), PromptMRG (Jin et al., 2024), CAMANet (Wang et al., 2024), DDaTR (Song et al., 2025), TGRG (Li et al., 2025), and REVTAF (Zhou et al., 2025). For the MIMIC-CXR dataset, as shown in Table 1, CAME-Grad achieves substantial and consistent performance improvements across all baseline models. Specifically, it achieves an average absolute improvement of 2.3% in CE metrics, while the NLG metrics remain stable. As emphasized by Song et al. (2025), since NLG scores are sensitive to stylistic variations, excelling in CE metrics that directly reflect diagnostic accuracy is more indicative of a model's clinical utility. Notably, on the strongest baseline REVTAF, we increase the F1-score from 0.589 to 0.607, reaching state-of-the-art performance. Table 2 illustrates the performance on the IU X-Ray dataset where CAME-Grad achieves a 1.9% average CE improvement while exhibiting robustness against data scarcity and gradient noise. Although domain shift and stylistic variances lead to more modest gains compared to MIMIC-CXR, our approach still improves the primary clinical efficacy of cross-

domain baselines such as PromptMRG and REVTAF under zero-shot settings. Furthermore, CAME-Grad delivers more substantial improvements for in-domain fine-tuning models like WCL and DCL by effectively resolving the "Double Dilemma" to release model potential during target-domain training. Finally, to rigorously confirm that CAME-Grad preserves the underlying clinical supervisory signals without degrading the diagnostic anchors, we present the detailed performance evaluation of the auxiliary classification task in Appendix F.

### 4.4.2. COMPARISON WITH MTL OPTIMIZERS

To further validate the superiority of our optimization strategy, we compare CAME-Grad with seven multi-task optimizers, including UW (Kendall et al., 2018), GradNorm (Chen et al., 2018), CAGrad (Liu et al., 2021), RotoGrad (Javaloy & Valera, 2022), FAMO (Liu et al., 2023), MM-Pareto (Wei & Hu, 2024), and STGU (Jeong & Yoon, 2025). All methods are implemented on the strongest REVTAF baseline and evaluated on the MIMIC-CXR dataset using their default hyperparameters. As Table 3 shows, while traditional methods demonstrate incremental improvements, their gains are constrained by specific failure modes. Magnitude-weighting methods like UW, GradNorm, and FAMO merely scale loss weights, rendering them powerless against destructive directional conflicts exceeding 90°. Meanwhile, globally-applied strict Pareto-based approaches like MMPareto suffer a performance collapse with average clinical efficacy plunging to 0.076. Minimizing the global gradient norm across non-shared parameters severely penal-

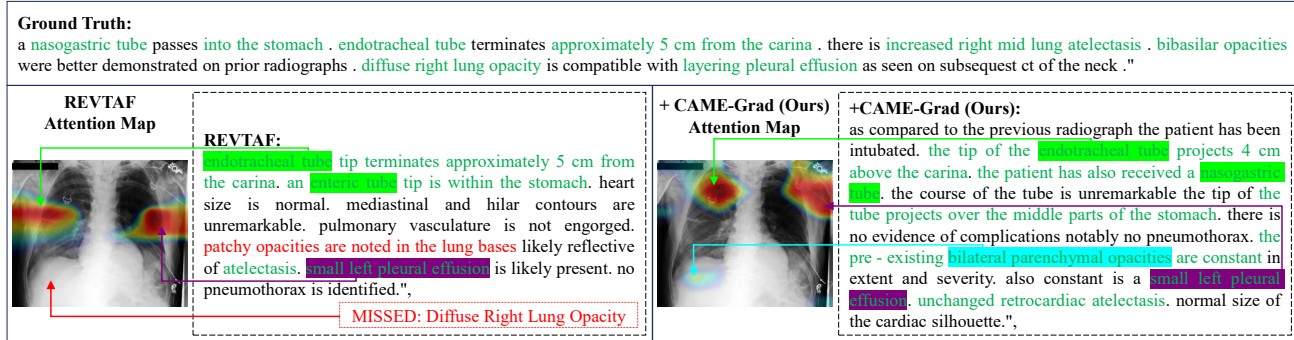

*Figure 4.* Qualitative comparison on the MIMIC-CXR test set. The Ground Truth report is shown at the top. The generated results of REVTAF and CAME-Grad are shown on the left and right, respectively. Green text indicates accurate clinical observations consistent with the Ground Truth, while red text indicates factual errors or missed diagnoses. Various colored highlights and arrows (e.g., green, cyan, purple) illustrate the correspondence between visual attention regions and specific descriptions in the generated text.

*Table 3.* Performance comparison with multi-task optimizers on the MIMIC-CXR test set. All methods are evaluated using REVTAF as the strong baseline. The best and second-best results in each metric are highlighted in bold and underlined, respectively, and ↑ indicates higher is better. CE Average denotes the arithmetic mean of Precision, Recall, and F1-score.

| REVTAF | PUBLICATION | CE METRICS | | | NLG METRICS | | | | CE AVG. ↑ |
|---|---|---|---|---|---|---|---|---|---|
| | | PREC. ↑ | REC. ↑ | F1 ↑ | B-1 ↑ | B-4 ↑ | MTR ↑ | R-L ↑ | |
| LINEAR | ICCV'25 | 0.615 | 0.617 | 0.589 | 0.465 | 0.184 | 0.199 | 0.336 | 0.607 |
| + UW | CVPR'18 | 0.619 | 0.622 | 0.593 | 0.457 | 0.178 | 0.196 | 0.333 | 0.611 |
| + GRADNORM | ICML'18 | 0.615 | 0.612 | 0.586 | 0.466 | 0.185 | 0.199 | 0.336 | 0.604 |
| + CAGRAD | NEURIPS'21 | **0.636** | 0.613 | 0.596 | 0.467 | 0.186 | 0.198 | 0.335 | 0.615 |
| + ROTOGRAD | ICLR'22 | 0.620 | 0.614 | 0.589 | **0.468** | **0.187** | 0.199 | **0.337** | 0.608 |
| + FAMO | NEURIPS'23 | 0.619 | 0.631 | 0.597 | 0.465 | 0.186 | **0.201** | 0.336 | 0.616 |
| + MMPARETO | ICML'24 | 0.071 | 0.085 | 0.073 | 0.010 | 0.000 | 0.004 | 0.016 | 0.076 |
| + STGU | ICLR'25 | 0.622 | 0.633 | 0.600 | 0.459 | 0.181 | 0.198 | 0.334 | 0.618 |
| **+ CAME-GRAD** | - | 0.633 | **0.637** | **0.607** | 0.466 | 0.185 | 0.200 | 0.336 | **0.626** |

izes the primary generative task, causing language modeling failure. Furthermore, while direction-projection methods like CAGrad successfully mitigate conflicts to achieve 0.636 precision, their rigid geometric projections restrict stochastic exploration. This triggers diffusion decay, trapping the model in sharp minima and degrading recall to 0.613, while gradient homogenization (e.g., RotoGrad) overly favors language fluency at the cost of diluting sharp diagnostic features. Ultimately, through its three cascaded stages, CAME-Grad simultaneously rectifies directional conflicts and compensates for kinetic energy loss to achieve the highest overall average clinical efficacy of 0.626 while maintaining stable report generation quality.

### 4.5. Analysis of CAME-Grad

#### 4.5.1. ABLATION STUDY

We conduct ablation studies on the MIMIC-CXR dataset using PromptMRG and DDaTR as representative models, as presented in Table 4 (full results in Appendices D and E). These results verify the contribution of the three stages of CAME-Grad. First, the results on PromptMRG confirm the

necessity of resolving direction conflicts. Enhancing magnitude without the direction guidance of S1 causes the F1-score to drop from the baseline of 0.482 to 0.476. This suggests that blind energy injection harms model performance without proper geometric direction guidance. Second, the results on DDaTR verify the importance of restoring and enhancing magnitude. Removing S2 leads to a significant decrease in F1-score from 0.438 of the full CAME-Grad to 0.425, indicating that compensating for projection-induced decay is essential to help the model escape local minima. Finally, removing the adaptive gradient fusion mechanism S3 results in a decrease in precision, while some NLG metrics degrade (e.g., BLEU-1) compared to the full CAME-Grad. This implies that S3 acts as a regularizer by introducing the inductive bias of the original gradient. It effectively balances aggressive exploration momentum with language smoothness, thereby improving overall diagnostic precision.

#### 4.5.2. QUALITATIVE RESULTS

To intuitively evaluate the clinical accuracy and coherence of generated reports, Figure 4 compares the qualitative re-

*Table 4.* Ablation study on the MIMIC-CXR dataset using PromptMRG and DDaTR as representative backbones. Best results are highlighted in bold and ↑ indicates higher is better. We compare the baseline linear scalarization with ablated variants to verify the contribution of each stage where S1, S2, and S3 represent Conflict-Averse Direction Rectification, Magnitude-Enhanced Energy Injection, and Adaptive Gradient Fusion, respectively.

| Base Model | Components | | | CE Metrics | | | NLG Metrics | | | | | | CE Avg. ↑ |
|---|---|---|---|---|---|---|---|---|---|---|---|---|---|
| | S1 | S2 | S3 | Prec. ↑ | Rec. ↑ | F1 ↑ | B-1 ↑ | B-2 ↑ | B-3 ↑ | B-4 ↑ | MTR ↑ | R-L ↑ | |
| PromptMRG | × | × | × | 0.505 | 0.518 | 0.482 | **0.396** | **0.236** | 0.152 | **0.105** | 0.155 | 0.265 | 0.502 |
| | × | ✓ | ✓ | 0.500 | 0.511 | 0.476 | 0.393 | 0.235 | 0.152 | **0.105** | 0.156 | **0.266** | 0.496 |
| | ✓ | × | ✓ | 0.513 | 0.520 | 0.487 | 0.393 | 0.235 | 0.152 | **0.105** | 0.156 | 0.265 | 0.507 |
| | ✓ | ✓ | × | 0.508 | **0.531** | 0.489 | 0.395 | **0.236** | **0.153** | **0.105** | 0.156 | **0.266** | 0.509 |
| | ✓ | ✓ | ✓ | **0.514** | 0.530 | **0.491** | **0.396** | **0.236** | **0.153** | **0.105** | **0.157** | **0.266** | **0.512** |
| DDaTR | × | × | × | 0.408 | 0.444 | 0.418 | 0.393 | 0.235 | 0.152 | 0.106 | 0.158 | 0.267 | 0.423 |
| | × | ✓ | ✓ | **0.439** | 0.452 | 0.436 | 0.396 | 0.240 | 0.158 | 0.111 | 0.160 | 0.271 | 0.442 |
| | ✓ | × | ✓ | 0.434 | 0.446 | 0.425 | 0.403 | 0.244 | 0.160 | 0.111 | 0.157 | 0.270 | 0.435 |
| | ✓ | ✓ | × | 0.425 | **0.467** | 0.435 | 0.402 | 0.244 | 0.160 | **0.113** | **0.162** | 0.272 | 0.442 |
| | ✓ | ✓ | ✓ | 0.432 | 0.463 | **0.438** | **0.405** | **0.245** | **0.161** | 0.113 | 0.161 | **0.273** | **0.444** |

sults of REVTAF and the CAME-Grad equipped REVTAF on the MIMIC-CXR test set. As shown in the figure, the ground truth clearly indicates diffuse right lung opacity and increased right mid lung atelectasis. However, REVTAF exhibits clinical limitations where it correctly identifies support devices by relying on high-frequency features in the upper mediastinum but fails to attend to the lung bases and misses the right-sided pathology. In contrast, the report generated by CAME-Grad aligns highly with the ground truth. It not only accurately localizes the endotracheal tube but also captures the global context to diagnose bilateral parenchymal opacities. The attention map confirms that the model successfully attends to the right lung base highlighted in cyan which is a critical region missed by the baseline model. This demonstrates that CAME-Grad successfully guides the model to focus on rare yet critical pathological features in the image by rectifying gradient direction and enhancing exploration momentum. Consequently, it significantly reduces the risk of clinical missed diagnoses and greatly improves the diagnostic credibility of the reports. More results are provided in Appendix G.

## 5. Conclusion

This study establishes the critical role of gradient dynamics in optimizing multi-task RRG. Our analysis reveals that the failure of traditional linear scalarization is rooted in its inability to decouple destructive gradient interference, thereby inducing the "Double Dilemma" of drift term deviation and diffusion term decay. CAME-Grad successfully overcomes this by constructing a coupling mechanism to reshape the optimization dynamics, thereby effectively reconciling the intrinsic conflict between report generation and clinical constraints. Experiments on the MIMIC-CXR and IU X-Ray datasets across eight baseline models demonstrate that CAME-Grad improves clinical efficacy by an average of 2.3% and 1.9%, respectively. Future work will focus on

extending this SDE-based gradient dynamics optimization algorithm to a broader range of multi-task medical imaging report generation tasks, providing theoretically deeper solutions to break the optimization bottlenecks in multi-task learning.

## Acknowledgements

This work was supported in part by the National Natural Science Foundation of China under Grant 62472368, Grant 62302427, and Grant 62462060, in part by the National Key Research and Development Program of China under Grant 2025YFF0515600.

## Impact Statement

This paper advances Machine Learning in automated radiology report generation (RRG). While the proposed CAME-Grad optimizer demonstrates potential in improving diagnostic efficiency and alleviating radiologist workload, deploying such models without rigorous clinical oversight carries risks of misdiagnosis and automation bias.

We explicitly acknowledge several limitations. First, clinical efficacy is evaluated via automatic labelers (e.g., CheXbert) and lexical metrics rather than direct clinician judgment. Consequently, real-world deployment requires stronger clinical safety validations. Second, our method introduces multiple hyperparameters, adding a tuning burden. Specifically, under extreme data scarcity and gradient noise (e.g., the IU X-Ray dataset), CAME-Grad guarantees a mathematically safe lower bound but requires careful tuning to reach the performance upper bound. Finally, our theoretical interpretation of the SDE dynamics currently exceeds direct empirical validation. Bridging this gap in real-world clinical trials remains a critical direction for future work to safely translate these theoretical gains into tangible clinical benefits.

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

## A. Universality Analysis of Internal Conflicts

In Section 3.2.1 of the main paper, we visualized the gradient dynamics using the REVTAF backbone to illustrate the intrinsic conflict in multi-task RRG. To further demonstrate that this "Double Dilemma" is a universal challenge inherent to the task paradigm rather than a model-specific issue, we extend this analysis to another representative baseline, PromptMRG. As illustrated in Figure A.1, the gradient dynamics of PromptMRG exhibit a striking resemblance to those of REVTAF shown in the main text. Specifically, PromptMRG exhibits a negative cosine similarity ratio of 49.8%, which is highly consistent with the 53.8% conflict ratio observed in REVTAF. This implies that regardless of the architecture, the intrinsic conflict between report generation and clinical constraints persists, causing the optimization direction to oscillate frequently. Furthermore, similar to the main experiments, the mean cosine similarity here is only $2.7 \times 10^{-3}$, indicating that the effective gradient magnitude is drastically reduced due to cancellations. This empirical evidence verifies that the negative interaction term $\mathcal{I}_k \ll 0$ is a widespread bottleneck in existing multi-task RRG methods, further justifying the necessity of our proposed CAME-Grad optimizer.

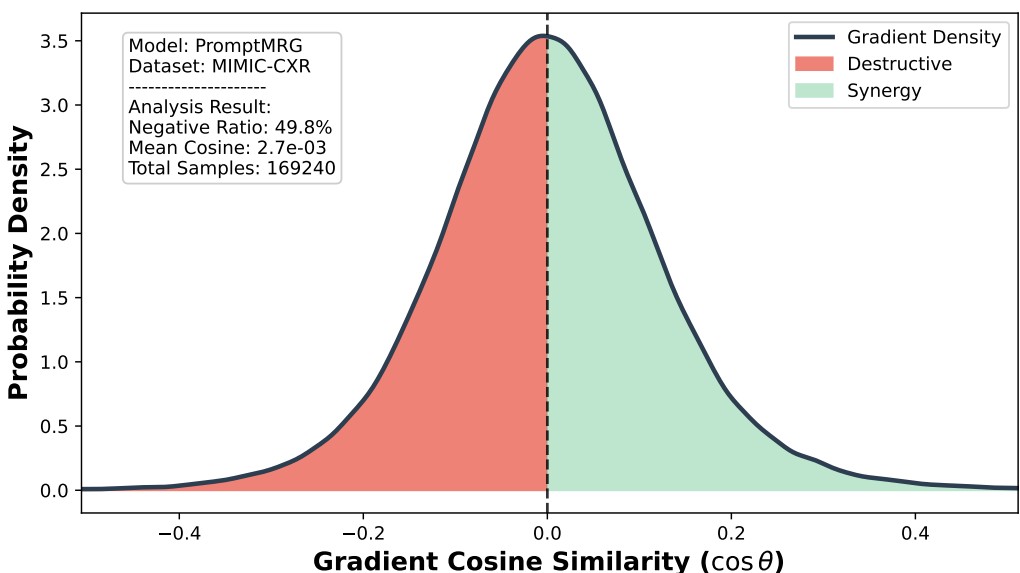

*Figure A.1.* Visualization of Gradient Conflict in PromptMRG. We analyze the distribution of gradient cosine similarities accumulated over 10 epochs on the MIMIC-CXR dataset. The x-axis represents the cosine similarity ($\cos \theta$) between gradients. The red area indicates destructive interference (negative similarity), where gradients cancel each other out, while the green area indicates synergy. Quantitatively, the Negative Ratio is 49.8%, meaning that nearly half of the gradient updates are conflicting during the training process. The mean cosine similarity is near zero ($2.7 \times 10^{-3}$), suggesting significant optimization instability.

## B. Extended Analysis on SDE Dynamics

### B.1. Geometric Validity of Drift Correction

To theoretically address the drift bias within our mechanistic hypothesis framework, we construct a minimax optimization in the tangent space:

$$\max_{\mathbf{u}} \min_{i \in \{0,\dots,K\}} \langle \mathbf{g}_i, \mathbf{u} \rangle \quad \text{s.t.} \quad \|\mathbf{u} - \boldsymbol{\mu}\| \le \rho \|\boldsymbol{\mu}\|, \tag{15}$$

where $\mathbf{g}_i$ denotes the gradient of the $i$-th task. By solving this optimization, the rectified direction $\mathbf{u}^*_{rect}$ is explicitly optimized to maximize the common descent alignment among all tasks. Within the continuous-time SDE dynamics, the expected parameter drift rate is defined as:

$$\frac{\mathbb{E}[d\Theta_t]}{dt} = -\mathbf{u}^*_{rect}. \tag{16}$$

This construction seeks to ensure that for each task, the first-order contribution to the local loss evolution is minimized or kept non-positive:

$$\langle \nabla_\Theta \mathcal{L}_i(\Theta_t), \frac{\mathbb{E}[d\Theta_t]}{dt} \rangle = -\langle \mathbf{g}_i, \mathbf{u}^*_{rect} \rangle \leq 0. \tag{17}$$

This provides a grounded geometric mechanism to suppress directional conflicts within the trust region, thereby mitigating the drift deviation suggested by our hypothesis.

### B.2. Dynamical Compensation and Covariance Measurements

As discussed in Section 3.3, Stage 2 compensates for the diffusion term decay by restoring and enhancing the gradient magnitude. In discrete mini-batch SGD, the parameter update along the projected direction involves implicit gradient noise $\hat{\mathbf{u}}^*_{rect} \sim \mathcal{N}(\mathbf{u}^*_{rect}, \Sigma_{rect})$. With amplification $\kappa$, the update becomes $\Delta\Theta = -\eta\kappa\hat{\mathbf{u}}^*_{rect}$. The expected drift $\mathbb{E}[\Delta\Theta] = -\eta\kappa\mathbf{u}^*_{rect}$ strictly preserves Stage 1's Pareto validity. Meanwhile, amplifying the update magnitude by $\kappa$ is dynamically equivalent to applying a larger local effective learning rate $\eta_{eff} = \eta\kappa$ along this 1D safe trajectory.

As established in classical SGD theory (Mandt et al., 2017), increasing the effective learning rate directly amplifies the stationary covariance of implicit noise. Mathematically, since the noise covariance is proportional to the outer product of gradients:

$$\Sigma(\Theta) \propto \mathbf{g}\mathbf{g}^\top, \tag{18}$$

scaling the gradient by a factor $\kappa$ results in the covariance trace growing by a factor of $\kappa^2$. Thus, CAME-Grad avoids adding unconstrained isotropic noise (which, as shown in Setting B below, destroys the clinical manifold). Instead, we use this directional scaling to amplify anisotropic exploration noise, compensating for the momentum decay caused by Stage 1's restricted projection.

To empirically verify this theoretical compensation, we monitor the evolution of the SGD noise covariance trace $\text{Tr}(\Sigma)$ during the training process. For the baseline (REVTAF with linear scalarization) on the MIMIC-CXR dataset, $\text{Tr}(\Sigma)$ remains at a low level of approximately 0.3215. In contrast, CAME-Grad significantly increases this physical quantity to approximately 4.4886 at convergence. This 14-fold increase in the physical measurement confirms that Stage 2 substantially enlarges the noise scale of the SDE, providing the necessary dynamical compensation to restore exploration kinetic energy.

### B.3. Comparative Analysis of Diffusion Mechanisms

To verify the necessity of magnitude enhancement strictly along the optimized direction, we conduct a comparative analysis between CAME-Grad and isotropic noise injection (Setting B: SGLD). As shown in Table B.1, while Setting A (omitting Stage 2) improves over the baseline via drift correction, it remains suboptimal due to energy depletion.

Crucially, Setting B's drop in clinical efficacy performance reveals the inherent difficulty of unconstrained high-dimensional optimization. Although isotropic noise $\mathcal{N}(0, \sigma^2 \mathbf{I})$ supplements the diffusion scale, it is highly likely to be orthogonal to the true gradient manifold. This directionless noise destroys the Stage 1 Pareto projection $\mathbf{u}^*_{rect}$, causing the trajectory to deviate from the clinical semantic manifold. Conversely, CAME-Grad restores the diffusion scale by scaling the magnitude along the rectified direction, maintaining both geometric validity and exploration momentum.

*Table B.1.* Comprehensive ablation study of diffusion mechanisms on REVTAF (MIMIC-CXR). Setting B utilizes Stochastic Gradient Langevin Dynamics (SGLD) to inject isotropic noise.

| Methods | Ablation | CE Metrics | | | NLG Metrics | | | | | | CE Avg. ↑ |
| --- | --- | --- | --- | --- | --- | --- | --- | --- | --- | --- | --- |
| | | Prec. ↑ | Rec. ↑ | F1 ↑ | B-1 ↑ | B-2 ↑ | B-3 ↑ | B-4 ↑ | MTR ↑ | R-L ↑ | |
| REVTAF (Linear) | - | 0.615 | 0.617 | 0.589 | 0.465 | 0.319 | 0.236 | 0.184 | 0.199 | 0.336 | 0.607 |
| Setting A | S1+S3 | 0.627 | 0.632 | 0.601 | **0.466** | **0.320** | **0.237** | **0.186** | **0.201** | **0.337** | 0.620 |
| Setting B | S1+SGLD+S3 | 0.610 | 0.585 | 0.569 | **0.466** | 0.319 | 0.236 | 0.184 | 0.196 | 0.332 | 0.588 |
| **CAME-Grad (Ours)** | S1+S2+S3 | **0.633** | **0.637** | **0.607** | **0.466** | **0.320** | **0.237** | 0.185 | 0.200 | 0.336 | **0.626** |

## C. Glossary of Multi-Task Optimization Terminology

To provide a comprehensive background on the multi-task optimization landscape and elaborate on the technical details of CAME-Grad, we outline the core terminologies, mathematical assumptions, and implementation mechanisms discussed in

this paper.

**Linear Scalarization:** The default and most prevalent baseline approach in multi-task learning (MTL). It formulates the global optimization objective as a static or dynamically weighted linear combination of individual task losses (i.e., $\mathcal{L}_{joint} = \sum_{i=0}^{K} \omega_i \mathcal{L}_i$). While computationally efficient, it inherently fails to address directional conflicts between task gradients.

**Geometric Validity:** A strict condition in gradient-based multi-task optimization requiring that the updated global gradient does not conflict with any individual task's descent direction. Geometrically, it mandates that the inner product between the rectified update vector $\mathbf{u}^*_{rect}$ and each task gradient $\mathbf{g}_i$ is non-negative (i.e., $\langle \mathbf{u}^*_{rect}, \mathbf{g}_i \rangle \geq 0$, meaning the angle does not exceed $90°$).

**Diffusion Kinetic Energy:** Viewed through the lens of Stochastic Differential Equations (SDEs), this term represents the inherent stochastic noise and exploration capability injected by mini-batch SGD during training. In the complex loss landscape of radiology report generation, maintaining sufficient diffusion kinetic energy is essential for the model to escape sharp, suboptimal local minima.

**Dual Variables ($\boldsymbol{\alpha}$):** In gradient manipulation methods, dual variables $\boldsymbol{\alpha} \in \Delta^{K+1}$ (the $(K+1)$-dimensional unit simplex) are used to represent the optimal combination weights. They allow the transformation of an intractable high-dimensional conflict resolution problem in the parameter space into a solvable convex optimization problem.

**Trust Region Assumption:** A foundational mathematical constraint in our optimization framework. It explicitly assumes that the corrected update direction $\mathbf{u}$ must reside within a bounded neighborhood of the mean gradient $\boldsymbol{\mu}$. This ensures that the rectified update preserves the underlying inductive bias and guarantees stable global convergence.

**Tensorized GPU Acceleration:** An implementation mechanism designed to overcome the computational bottleneck of high-dimensional gradient manipulations. Instead of transferring parameters to the CPU, CAME-Grad solves the tensorized dual problem and executes the rectification stages directly on the GPU via parallel operations, ensuring minimal training overhead.

**Min-Max Trust-Region Formulation:** The core theoretical outcome of the min-max formula Eq. 8 in Stage 1 is providing a mathematical guarantee for strict local Pareto descent. Specifically, the inner *min* operator locates the most easily sacrificed task, while the outer *max* operator maximizes the descent benefit of this worst task. Incorporating the trust region constraint forces the generation of a consensus direction possessing a positive inner product with all task gradients. This mathematically establishes a safe lower bound, ensuring the model never destroys the generation manifold while capturing clinical features, thus eliminating negative transfer.

## D. Comprehensive Analysis on MIMIC-CXR

In this section, we provide a comprehensive analysis of hyperparameter sensitivity and present the full quantitative results on the MIMIC-CXR dataset. All models were implemented following the unified experimental setup described in Section 4.3 of the main paper, ensuring a fair comparison across backbone architectures.

### D.1. Hyperparameter Sensitivity

Table D.1 summarizes the optimal hyperparameter configurations for the eight baseline models. We observe a strong correlation between the optimal settings and the rigidity of the auxiliary tasks employed by each baseline, revealing distinct optimization behaviors across different constraint types.

Models employing rigid, discrete auxiliary tasks, specifically PromptMRG, REVTAF, DDaTR, and DCL, consistently require a lower adaptive fusion coefficient ($\nu \in [0.1, 0.2]$) combined with standard energy injection ($\kappa = 1.5$). This configuration is attributed to the severe gradient direction deviation caused by their respective auxiliary objectives, such as multi-label classification or retrieval-based alignment. As demonstrated in our gradient analysis, these hard constraints often induce gradients that are nearly orthogonal or opposing to the report generation manifold. Empirically, we observed that increasing $\nu$ beyond 0.3 for these models leads to a resurgence of conflict issues, manifesting as unstable convergence or regression to baseline performance. Consequently, a lower $\nu$ effectively down-weights the original joint gradient, forcing the optimization to rely more heavily on the rectified direction $\mathbf{u}^*_{rect}$ computed in Stage 1 to mitigate destructive interference.

In contrast, models relying on geometric or prototype-based constraints, such as XProNet and CAMANet, benefit from a

*Table D.1.* Detailed hyperparameter settings of CAME-Grad for different baselines on MIMIC-CXR. The settings correlate with the rigidity of auxiliary tasks: models with hard constraints (e.g., Classification) generally require lower $\nu$ to rely more on rectified gradients, while soft constraints allow higher $\nu$.

| Model | Auxiliary Task Type | $\rho$ | $\kappa$ | $\nu$ |
|---|---|---|---|---|
| *High Conflict / Rigid Constraints* | | | | |
| PromptMRG | Multi-label Classification | 0.2 | 1.5 | 0.1 |
| REVTAF | Retrieval & Alignment | 0.5 | 1.5 | 0.2 |
| DDaTR | Longitudinal Diff. | 0.5 | 1.5 | 0.2 |
| DCL | Graph Node Classif. | 0.5 | 1.5 | 0.2 |
| *Geometric / Prototype Constraints* | | | | |
| XProNet | Prototype Matching | 0.3 | 1.5 | 0.4 |
| CAMANet | CAM-guided Attention | 0.4 | 1.5 | 0.5 |
| *Soft / Weak Constraints* | | | | |
| TGRG | Topic Modeling | 0.1 | 1.7 | 0.8 |
| WCL | Weakly-supervised Contrastive | 0.2 | 1.0 | 0.9 |

*Table D.2.* Complete performance comparison on the MIMIC-CXR test set with specific hyperparameter configurations. Best results for each baseline pair are highlighted in bold. The inclusion of $\rho, \kappa, \nu$ demonstrates the specific tuning required for each architecture.

| Model | Publication | Hyperparams | | | CE Metrics | | | NLG Metrics | | | | | | CE Avg. ↑ |
|---|---|---|---|---|---|---|---|---|---|---|---|---|---|---|
| | | $\rho$ | $\kappa$ | $\nu$ | Prec. ↑ | Rec. ↑ | F1 ↑ | B-1 ↑ | B-2 ↑ | B-3 ↑ | B-4 ↑ | MTR ↑ | R-L ↑ | |
| WCL | EMNLP'21 | | | | 0.382 | 0.298 | 0.312 | 0.329 | 0.203 | 0.136 | 0.098 | 0.135 | 0.277 | 0.331 |
| + CAME-Grad | | 0.2 | 1.0 | 0.9 | **0.424** | **0.324** | **0.344** | **0.349** | **0.219** | **0.150** | **0.108** | **0.142** | **0.284** | **0.364** |
| XProNet | ECCV'22 | | | | 0.382 | 0.304 | 0.314 | 0.325 | 0.202 | 0.137 | **0.099** | 0.134 | 0.260 | 0.333 |
| + CAME-Grad | | 0.3 | 1.5 | 0.4 | **0.391** | **0.309** | **0.322** | **0.329** | **0.204** | **0.138** | 0.099 | **0.135** | **0.261** | **0.341** |
| DCL | CVPR'23 | | | | 0.265 | 0.246 | 0.236 | 0.234 | 0.136 | 0.087 | 0.061 | 0.108 | 0.211 | 0.249 |
| + CAME-Grad | | 0.5 | 1.5 | 0.2 | **0.319** | **0.298** | **0.286** | **0.280** | **0.162** | **0.104** | **0.074** | **0.122** | **0.220** | **0.301** |
| PromptMRG | AAAI'24 | | | | 0.505 | 0.518 | 0.482 | **0.396** | **0.236** | 0.152 | **0.105** | 0.155 | 0.265 | 0.502 |
| + CAME-Grad | | 0.2 | 1.5 | 0.1 | **0.514** | **0.530** | **0.491** | **0.396** | **0.236** | **0.153** | **0.105** | **0.157** | **0.266** | **0.512** |
| CAMANet | JBHI'24 | | | | 0.405 | 0.303 | 0.321 | **0.327** | 0.202 | 0.136 | 0.097 | 0.134 | 0.274 | 0.343 |
| + CAME-Grad | | 0.4 | 1.5 | 0.5 | **0.410** | **0.313** | **0.330** | **0.327** | **0.203** | **0.137** | **0.099** | **0.135** | **0.275** | **0.351** |
| DDaTR | TMI'25 | | | | 0.408 | 0.444 | 0.418 | 0.393 | 0.235 | 0.152 | 0.106 | 0.158 | 0.267 | 0.423 |
| + CAME-Grad | | 0.5 | 1.5 | 0.2 | **0.432** | **0.463** | **0.438** | **0.405** | **0.245** | **0.161** | **0.113** | **0.161** | **0.273** | **0.444** |
| TGRG | MedIA'25 | | | | 0.457 | 0.435 | 0.417 | 0.423 | 0.258 | 0.164 | 0.109 | 0.162 | 0.284 | 0.436 |
| + CAME-Grad | | 0.1 | 1.7 | 0.8 | **0.493** | **0.466** | **0.450** | **0.438** | **0.268** | **0.173** | **0.117** | **0.169** | **0.285** | **0.470** |
| REVTAF | ICCV'25 | | | | 0.615 | 0.617 | 0.589 | 0.465 | 0.319 | 0.236 | 0.184 | 0.199 | **0.336** | 0.607 |
| + CAME-Grad | | 0.5 | 1.5 | 0.2 | **0.633** | **0.637** | **0.607** | **0.466** | **0.320** | **0.237** | **0.185** | **0.200** | **0.336** | **0.626** |

balanced configuration with a medium fusion coefficient ($\nu \in [0.4, 0.5]$). While these methods utilize prototypes or attention maps that impose structural requirements on the feature space, their constraints are softer than hard labels. The moderate $\nu$ value indicates a necessary trade-off where the optimizer must rectify conflicting components while preserving the beneficial inductive biases, such as cluster structures provided by the prototypes. Notably, CAMANet requires a larger rectification radius ($\rho = 0.4$), which suggests that its attention-guided gradients exhibit a higher variance in angular distribution. A broader trust region is thus necessary to prevent the optimizer from over-correcting potentially useful attention shifts, balancing stability with flexibility.

Finally, for baselines incorporating loose or global semantic guidance, namely TGRG and WCL, we find that a higher fusion coefficient ($\nu \in [0.8, 0.9]$) yields optimal performance. The auxiliary tasks in these models, such as topic modeling or weakly-supervised contrastive learning, provide global semantic regularization rather than per-sample penalties. This results in a high degree of synergy between tasks, allowing the optimizer to trust the original joint gradient direction. For WCL specifically, the conflict is often merely stochastic noise rather than structural misalignment, justifying the high retention of the original gradient to maintain training efficiency. Interestingly, TGRG requires the highest energy gain ($\kappa = 1.7$), likely because topic distributions induce a flatter optimization landscape, requiring stronger kinetic energy to escape local minima.

*Table D.3.* Comprehensive ablation study of CAME-Grad across eight baseline architectures on MIMIC-CXR, ordered by publication year. We strictly compare the Baseline (Linear Scalarization) with three ablated variants and the Full model. **S1**: Conflict-Averse Direction Rectification; **S2**: Magnitude-Enhanced Energy Injection; **S3**: Adaptive Gradient Fusion. The results demonstrate that CAME-Grad (Full) consistently achieves the best trade-off between clinical consistency and generation quality.

| Base Model | Components | | | CE Metrics | | | NLG Metrics | | | | | | CE Avg. ↑ |
| --- | --- | --- | --- | --- | --- | --- | --- | --- | --- | --- | --- | --- | --- |
| | S1 | S2 | S3 | Prec. ↑ | Rec. ↑ | F1 ↑ | B-1 ↑ | B-2 ↑ | B-3 ↑ | B-4 ↑ | MTR ↑ | R-L ↑ | |
| WCL | × | × | × | 0.382 | 0.298 | 0.312 | 0.329 | 0.203 | 0.136 | 0.098 | 0.135 | 0.277 | 0.331 |
| | × | ✓ | ✓ | 0.412 | 0.304 | 0.322 | 0.341 | 0.213 | 0.144 | 0.105 | 0.139 | 0.280 | 0.346 |
| | ✓ | × | ✓ | 0.412 | **0.331** | 0.341 | **0.352** | **0.220** | **0.150** | **0.108** | **0.142** | 0.282 | 0.361 |
| | ✓ | ✓ | × | 0.379 | 0.275 | 0.293 | 0.332 | 0.204 | 0.137 | 0.099 | 0.136 | 0.274 | 0.316 |
| | ✓ | ✓ | ✓ | **0.424** | 0.324 | **0.344** | 0.349 | 0.219 | **0.150** | **0.108** | **0.142** | **0.284** | **0.364** |
| XProNet | × | × | × | 0.382 | 0.304 | 0.314 | 0.325 | 0.202 | 0.137 | 0.099 | 0.134 | 0.260 | 0.333 |
| | × | ✓ | ✓ | 0.362 | 0.279 | 0.294 | **0.329** | **0.205** | **0.139** | **0.101** | 0.134 | **0.262** | 0.312 |
| | ✓ | × | ✓ | **0.392** | 0.307 | 0.321 | 0.323 | 0.200 | 0.135 | 0.098 | 0.134 | 0.259 | 0.340 |
| | ✓ | ✓ | × | 0.389 | 0.303 | 0.317 | 0.323 | 0.199 | 0.134 | 0.096 | 0.132 | 0.257 | 0.336 |
| | ✓ | ✓ | ✓ | 0.391 | **0.309** | **0.322** | **0.329** | 0.204 | 0.138 | 0.099 | **0.135** | 0.261 | **0.341** |
| DCL | × | × | × | 0.265 | 0.246 | 0.236 | 0.234 | 0.136 | 0.087 | 0.061 | 0.108 | 0.211 | 0.249 |
| | × | ✓ | ✓ | 0.288 | 0.262 | 0.255 | 0.281 | 0.162 | 0.105 | 0.074 | 0.122 | 0.222 | 0.268 |
| | ✓ | × | ✓ | 0.290 | 0.276 | 0.262 | 0.279 | 0.161 | 0.104 | 0.074 | 0.121 | 0.222 | 0.276 |
| | ✓ | ✓ | × | 0.313 | 0.291 | 0.281 | **0.294** | **0.170** | **0.110** | **0.077** | **0.125** | **0.223** | 0.295 |
| | ✓ | ✓ | ✓ | **0.319** | **0.298** | **0.286** | 0.280 | 0.162 | 0.104 | 0.074 | 0.122 | 0.220 | **0.301** |
| PromptMRG | × | × | × | 0.505 | 0.518 | 0.482 | **0.396** | **0.236** | 0.152 | **0.105** | 0.155 | 0.265 | 0.502 |
| | × | ✓ | ✓ | 0.500 | 0.511 | 0.476 | 0.393 | 0.235 | 0.152 | **0.105** | 0.156 | **0.266** | 0.496 |
| | ✓ | × | ✓ | 0.513 | 0.520 | 0.487 | 0.393 | 0.235 | 0.152 | **0.105** | 0.156 | 0.265 | 0.507 |
| | ✓ | ✓ | × | 0.508 | **0.531** | 0.489 | 0.395 | **0.236** | 0.153 | **0.105** | 0.156 | **0.266** | 0.509 |
| | ✓ | ✓ | ✓ | **0.514** | 0.530 | **0.491** | **0.396** | **0.236** | 0.153 | **0.105** | **0.157** | **0.266** | **0.512** |
| CAMANet | × | × | × | 0.405 | 0.303 | 0.321 | 0.327 | 0.202 | 0.136 | 0.097 | 0.134 | 0.274 | 0.343 |
| | × | ✓ | ✓ | 0.387 | 0.292 | 0.312 | **0.336** | **0.209** | **0.141** | **0.102** | **0.137** | **0.276** | 0.330 |
| | ✓ | × | ✓ | 0.395 | 0.314 | 0.326 | 0.328 | 0.203 | 0.137 | 0.099 | 0.135 | 0.273 | 0.345 |
| | ✓ | ✓ | × | 0.404 | **0.322** | **0.337** | 0.327 | 0.204 | 0.139 | 0.100 | 0.134 | **0.276** | **0.354** |
| | ✓ | ✓ | ✓ | **0.410** | 0.313 | 0.330 | 0.327 | 0.203 | 0.137 | 0.099 | 0.135 | 0.275 | 0.351 |
| DDATR | × | × | × | 0.408 | 0.444 | 0.418 | 0.393 | 0.235 | 0.152 | 0.106 | 0.158 | 0.267 | 0.423 |
| | × | ✓ | ✓ | **0.439** | 0.452 | 0.436 | 0.396 | 0.240 | 0.158 | 0.111 | 0.160 | 0.271 | 0.442 |
| | ✓ | × | ✓ | 0.434 | 0.446 | 0.425 | 0.403 | 0.244 | 0.160 | 0.111 | 0.157 | 0.270 | 0.435 |
| | ✓ | ✓ | × | 0.425 | **0.467** | 0.435 | 0.402 | 0.244 | 0.160 | **0.113** | **0.162** | 0.272 | 0.442 |
| | ✓ | ✓ | ✓ | 0.432 | 0.463 | **0.438** | **0.405** | **0.245** | **0.161** | **0.113** | 0.161 | **0.273** | **0.444** |
| TGRG | × | × | × | 0.457 | 0.435 | 0.417 | 0.423 | 0.258 | 0.164 | 0.109 | 0.162 | 0.284 | 0.436 |
| | × | ✓ | ✓ | 0.483 | **0.487** | **0.455** | **0.440** | 0.268 | 0.175 | 0.119 | **0.169** | 0.287 | **0.475** |
| | ✓ | × | ✓ | 0.483 | 0.467 | 0.445 | 0.436 | **0.269** | **0.176** | **0.121** | **0.169** | 0.290 | 0.465 |
| | ✓ | ✓ | × | 0.484 | 0.448 | 0.436 | 0.429 | 0.260 | 0.169 | 0.114 | 0.166 | 0.285 | 0.456 |
| | ✓ | ✓ | ✓ | **0.493** | 0.466 | 0.450 | 0.438 | 0.268 | 0.173 | 0.117 | **0.169** | 0.285 | 0.470 |
| REVTAF | × | × | × | 0.615 | 0.617 | 0.589 | 0.465 | 0.319 | 0.236 | 0.184 | 0.199 | 0.336 | 0.607 |
| | × | ✓ | ✓ | 0.624 | 0.630 | 0.599 | **0.468** | **0.322** | **0.239** | **0.187** | 0.201 | **0.338** | 0.618 |
| | ✓ | × | ✓ | 0.627 | 0.632 | 0.601 | 0.466 | 0.320 | 0.237 | 0.186 | 0.201 | 0.337 | 0.620 |
| | ✓ | ✓ | × | 0.628 | 0.630 | 0.601 | 0.467 | 0.321 | **0.239** | **0.187** | **0.202** | 0.337 | 0.620 |
| | ✓ | ✓ | ✓ | **0.633** | **0.637** | **0.607** | 0.466 | 0.320 | 0.237 | 0.185 | 0.200 | 0.336 | **0.626** |

## D.2. Performance Evaluation

Table D.2 presents the comprehensive performance comparison on the MIMIC-CXR test set. By explicitly listing the hyperparameter configuration $(\rho, \kappa, \nu)$ alongside Clinical Efficacy (CE) and Natural Language Generation (NLG) metrics, we demonstrate how specific tuning adapts to different backbone architectures to achieve consistent improvements.

## D.3. Ablation Study for Component

In this subsection, we provide a detailed component analysis of CAME-Grad across eight baseline architectures. To rigorously evaluate the contribution of each algorithmic stage, we strictly compare the Full CAME-Grad model against the baseline (Linear Scalarization) and three variants, each removing one key stage: Stage 1 (Conflict-Averse Direction Rectification), Stage 2 (Magnitude-Enhanced Energy Injection), and Stage 3 (Adaptive Gradient Fusion).

**Stage 1: Conflict-Averse Direction Rectification.** The contribution of conflict rectification in Stage 1 is most pronounced in models employing rigid auxiliary constraints, such as PromptMRG and REVTAF. These models introduce hard constraints—multi-label classification or retrieval alignment—that often generate gradients orthogonal or even opposing to the primary generation objective. For PromptMRG, removing Stage 1 results in a significant drop in F1-score from 0.491 to 0.476, alongside a notable decrease in Precision. This validates our hypothesis that without manifold-guided rectification, the sharp gradient conflicts stemming from hard labels destructively interfere with the language modeling manifold. Similarly, for REVTAF, the removal of Stage 1 leads to a regression in clinical efficacy metrics, suggesting that projecting auxiliary gradients onto a tangent space is essential for transforming conflicting signals into beneficial guidance.

**Stage 2: Magnitude-Enhanced Energy Injection.** Stage 2 acts as a universal stabilizer by restoring and enhancing the gradient magnitude after rectification. While Stage 1 ensures the update direction is correct, the rectification process (projection) often reduces the vector norm, potentially causing a loss of kinetic energy during optimization. Our results on XProNet and DDaTR confirm this theoretical concern. For XProNet, removing Stage 2 leads to a consistent, albeit subtle, decline in both clinical efficacy (CE Average drops from 0.341 to 0.340) and NLG metrics compared to the full model. This suggests that without energy injection, the optimizer may suffer from "gradient vanishing" in the projected space, causing it to stall in saddle points or converge slower. Restoring and enhancing the magnitude ensures that the rectified update retains sufficient momentum to escape local minima.

**Stage 3: Adaptive Gradient Fusion.** Stage 3, governed by the adaptive fusion coefficient $\nu$, proves critical for models with complex or weak constraints, such as WCL and DCL. This stage dynamically balances the trust between the original joint gradient and the rectified update based on the current conflict level. WCL exhibits the highest sensitivity to this stage. Removing adaptive fusion causes a catastrophic performance drop, with the F1-score plummeting from 0.344 to 0.293. This confirms that for weakly-supervised tasks, a fixed weighting scheme fails to capture the time-varying synergy between contrastive learning and language modeling. Stage 3 allows the optimizer to dynamically "trust" the original gradient when synergy is high, which is essential for preserving the training efficiency of synergistic tasks. For DCL, while removing Stage 3 marginally improves NLG scores, it degrades clinical efficacy, indicating that adaptive fusion is key to steering the model towards clinically accurate terminology rather than just fluent text.

**Precision-Recall Trade-off.** It is also worth noting the trade-off behavior observed in models like TGRG and CAMANet. In some cases, specific ablated variants (e.g., without S1 or S3) achieve higher Recall, F1-scores, or lexical metrics than the full model. However, the full CAME-Grad configuration generally maintains highly competitive or superior Precision across most baselines. The full tri-stage mechanism effectively suppresses the generation of non-existent pathologies driven by noisy auxiliary signals. Therefore, while individual stages might occasionally be omitted to maximize specific recall-oriented metrics, the complete CAME-Grad framework provides the most robust and clinically reliable generation strategy across diverse constraint types.

## E. Generalizability and Robustness Analysis on IU X-Ray

In this section, we extend our evaluation to the IU X-Ray dataset to verify the generalizability of CAME-Grad across different data scales and domain distributions. We provide a detailed analysis of hyperparameter adaptation strategies in Table E.1, followed by a comprehensive performance comparison in Table E.2.

### E.1. Hyperparameter Sensitivity and Zero-Shot Settings

Consistent with our observations on MIMIC-CXR, the optimal hyperparameter settings on IU X-Ray exhibit a strong correlation with task rigidity. However, the distinct data scarcity of the IU X-Ray dataset and our experimental setup involving cross-domain evaluation introduce specific adaptation patterns. A key distinction in our experiments is the use of zero-shot inference for PromptMRG, REVTAF, and DDaTR. As these models utilize checkpoints pre-trained on MIMIC-CXR and are evaluated directly on IU X-Ray without fine-tuning, their hyperparameter configurations $(\rho, \kappa, \nu)$ are strictly inherited from the MIMIC-CXR training phase. The consistent improvements in overall clinical efficacy observed in Table E.2, such as PromptMRG achieving a 0.209 CE Average compared to the 0.202 baseline, demonstrate the intrinsic transferability of the optimization trajectory found by CAME-Grad. This suggests that the rectified gradient directions computed on the source domain successfully capture generalized geometric properties of the report generation manifold, which remains valid even when transferred to a target domain with different distribution characteristics.

*Table E.1.* Detailed hyperparameter settings of CAME-Grad for different baselines on IU X-Ray. For zero-shot models, parameters are inherited from MIMIC-CXR (marked as -). For supervised models, we observe a general trend towards higher $\nu$ values (e.g., DCL, XProNet) for regularization, with the notable exception of WCL, which requires a lower $\nu$ to suppress increased label noise in low-resource contrastive learning.

| Model | Auxiliary Task Type | $\rho$ | $\kappa$ | $\nu$ |
|---|---|---|---|---|
| *Zero-Shot Inference (Inherited Configs)* | | | | |
| PromptMRG | Multi-label Classification | - | - | - |
| REVTAF | Retrieval & Alignment | - | - | - |
| DDaTR | Longitudinal Diff. | - | - | - |
| *Supervised Training (High Regularization)* | | | | |
| DCL | Graph Node Classif. | 0.5 | 1.5 | 0.8 |
| XProNet | Prototype Matching | 0.1 | 1.0 | 0.8 |
| CAMANet | CAM-guided Attention | 0.5 | 2.0 | 0.9 |
| TGRG | Topic Modeling | 0.1 | 1.0 | 0.9 |
| *Supervised Training (Others)* | | | | |
| WCL | Weakly-supervised Contrastive | 0.1 | 1.1 | 0.4 |

*Table E.2.* Complete performance comparison on the IU X-Ray test set. $*$ indicates zero-shot inference, where models are evaluated directly using MIMIC-CXR checkpoints; thus, their hyperparameters are inherited rather than tuned on IU X-Ray. † denotes standard supervised training. Best results are in bold.

| MODEL | PUBLICATION | HYPERPARAMS | | | CE METRICS | | | NLG METRICS | | | | | | CE AVG. ↑ |
|---|---|---|---|---|---|---|---|---|---|---|---|---|---|---|
| | | $\rho$ | $\kappa$ | $\nu$ | PREC. ↑ | REC. ↑ | F1 ↑ | B-1 ↑ | B-2 ↑ | B-3 ↑ | B-4 ↑ | MTR ↑ | R-L ↑ | |
| WCL† | EMNLP'21 | | | | 0.490 | 0.498 | 0.491 | 0.325 | 0.195 | 0.137 | 0.101 | 0.137 | 0.318 | 0.493 |
| + CAME-GRAD | | 0.1 | 1.1 | 0.4 | **0.522** | **0.514** | **0.516** | **0.362** | **0.226** | **0.163** | **0.124** | **0.156** | **0.332** | **0.517** |
| XPRONET† | ECCV'22 | | | | 0.598 | 0.588 | 0.590 | 0.440 | 0.271 | 0.183 | 0.132 | **0.182** | **0.336** | 0.592 |
| + CAME-GRAD | | 0.1 | 1.0 | 0.8 | **0.619** | **0.603** | **0.606** | **0.463** | **0.292** | **0.204** | **0.151** | 0.181 | 0.333 | **0.609** |
| DCL† | CVPR'23 | | | | 0.527 | 0.525 | 0.525 | 0.393 | 0.255 | 0.186 | 0.144 | 0.185 | 0.298 | 0.526 |
| + CAME-GRAD | | 0.5 | 1.5 | 0.8 | **0.575** | **0.569** | **0.571** | **0.399** | **0.262** | **0.193** | **0.151** | **0.191** | **0.312** | **0.572** |
| PROMPTMRG* | AAAI'24 | | | | 0.200 | 0.210 | 0.197 | **0.419** | **0.248** | 0.158 | 0.106 | **0.161** | **0.314** | 0.202 |
| + CAME-GRAD | | - | - | - | **0.207** | **0.217** | **0.203** | **0.419** | **0.248** | **0.159** | **0.107** | 0.158 | 0.308 | **0.209** |
| CAMANET† | JBHI'24 | | | | 0.503 | 0.490 | 0.494 | 0.398 | 0.256 | 0.185 | 0.142 | 0.166 | 0.339 | 0.496 |
| + CAME-GRAD | | 0.5 | 2.0 | 0.9 | **0.520** | **0.514** | **0.516** | **0.409** | **0.261** | **0.190** | **0.145** | **0.170** | **0.349** | **0.517** |
| DDATR* | TMI'25 | | | | **0.308** | 0.282 | 0.257 | 0.428 | 0.253 | 0.160 | 0.107 | 0.161 | **0.314** | 0.282 |
| + CAME-GRAD | | - | - | - | 0.281 | **0.308** | **0.259** | **0.433** | **0.257** | **0.164** | **0.110** | **0.163** | 0.313 | **0.283** |
| TGRG† | MEDIA'25 | | | | 0.508 | 0.497 | 0.500 | 0.471 | 0.302 | 0.204 | **0.146** | 0.189 | **0.392** | 0.502 |
| + CAME-GRAD | | 0.1 | 1.0 | 0.9 | **0.527** | **0.530** | **0.525** | **0.498** | **0.308** | **0.207** | 0.145 | **0.194** | 0.382 | **0.527** |
| REVTAF* | ICCV'25 | | | | 0.292 | 0.291 | 0.283 | 0.420 | 0.249 | **0.159** | **0.106** | **0.178** | **0.311** | 0.289 |
| + CAME-GRAD | | - | - | - | **0.304** | **0.302** | **0.294** | **0.424** | **0.251** | **0.159** | 0.105 | 0.177 | 0.310 | **0.300** |

In contrast, for models trained from scratch on IU X-Ray, such as DCL, XProNet, CAMANet, and TGRG, we observe a significant shift towards higher fusion coefficients compared to their MIMIC-CXR counterparts. For instance, DCL requires $\nu = 0.2$ on MIMIC-CXR but favors $\nu = 0.8$ on IU X-Ray, and similarly, XProNet shifts from $\nu = 0.4$ to $\nu = 0.8$. This phenomenon is attributed to the necessity of regularization in low-resource scenarios. Since the IU X-Ray dataset contains significantly fewer samples, the generator is highly prone to overfitting the training data. In this context, auxiliary tasks such as graph node classification or prototype matching provide essential structural priors. A higher $\nu$ instructs the optimizer to retain more of the original gradient information derived from these tasks, effectively acting as a regularizer that prevents the model from collapsing into memorization. Interestingly, WCL is the only exception where the optimal $\nu$ decreases to 0.4 on IU X-Ray. This is attributed to the fact that weakly-supervised contrastive signals become significantly noisier in low-resource settings; thus, a lower $\nu$ is required to allow CAME-Grad to more aggressively rectify stochastic interference and maintain optimization stability.

## E.2. Quantitative Performance Comparison

Table E.2 presents the comprehensive performance comparison. CAME-Grad consistently improves overall clinical efficacy, as evidenced by the gains in CE Average across all baselines, despite minor trade-offs in specific NLG metrics for some

models. Notably, even for the supervised models that require strong regularization (high $\nu$), our method achieves substantial improvements (e.g., DCL improves by 4.6% in CE Average), validating that CAME-Grad effectively balances the trade-off between conflict resolution and structural regularization.

### E.3. Ablation Study for Component

In this subsection, we present a comprehensive component analysis of CAME-Grad on the IU X-Ray dataset. As detailed in Table E.3, we rigorously compare the full CAME-Grad optimizer against the baseline (Linear Scalarization) and three ablated variants. Unlike the MIMIC-CXR experiments, the IU X-Ray benchmark involves a complex mix of supervised training on small-scale data and zero-shot inference using cross-domain checkpoints. This unique experimental setup reveals distinct behavioral patterns in how the three stages of CAME-Grad contribute to generalization and robustness across domains.

*Table E.3.* Comprehensive ablation study of CAME-Grad across eight baseline architectures on IU X-Ray, ordered by publication year. ∗ and † denote zero-shot inference and supervised training, respectively. We strictly compare the Baseline (Linear Scalarization) with three ablated variants and the Full model. **S1**: Conflict-Averse Direction Rectification; **S2**: Magnitude-Enhanced Energy Injection; **S3**: Adaptive Gradient Fusion. CE Avg. denotes the average of Precision, Recall, and F1-score. The best results for each backbone are highlighted in bold.

| BASE MODEL | S1 | S2 | S3 | PREC. ↑ | REC. ↑ | F1 ↑ | B-1 ↑ | B-2 ↑ | B-3 ↑ | B-4 ↑ | MTR ↑ | R-L ↑ | CE AVG. ↑ |
|---|---|---|---|---|---|---|---|---|---|---|---|---|---|
| | | | | **CE METRICS** | | | **NLG METRICS** | | | | | | |
| WCL† | × | × | × | 0.490 | 0.498 | 0.491 | 0.325 | 0.195 | 0.137 | 0.101 | 0.137 | 0.318 | 0.493 |
| | × | ✓ | ✓ | 0.515 | 0.506 | 0.508 | 0.330 | 0.194 | 0.134 | 0.098 | 0.140 | 0.316 | 0.510 |
| | ✓ | × | ✓ | 0.505 | 0.499 | 0.501 | 0.297 | 0.177 | 0.124 | 0.092 | 0.134 | 0.311 | 0.502 |
| | ✓ | ✓ | × | 0.512 | 0.505 | 0.506 | 0.298 | 0.182 | 0.129 | 0.097 | 0.136 | 0.317 | 0.508 |
| | ✓ | ✓ | ✓ | **0.522** | **0.514** | **0.516** | **0.362** | **0.226** | **0.163** | **0.124** | **0.156** | **0.332** | **0.517** |
| XPRONET† | × | × | × | 0.598 | 0.588 | 0.590 | 0.440 | 0.271 | 0.183 | 0.132 | 0.182 | 0.336 | 0.592 |
| | × | ✓ | ✓ | 0.610 | 0.592 | 0.597 | **0.491** | **0.315** | **0.219** | **0.162** | **0.200** | **0.348** | 0.600 |
| | ✓ | × | ✓ | 0.589 | 0.572 | 0.577 | 0.457 | 0.282 | 0.198 | 0.148 | 0.180 | 0.327 | 0.579 |
| | ✓ | ✓ | × | 0.564 | 0.555 | 0.558 | 0.442 | 0.269 | 0.190 | 0.144 | 0.181 | 0.327 | 0.559 |
| | ✓ | ✓ | ✓ | **0.619** | **0.603** | **0.606** | 0.463 | 0.292 | 0.204 | 0.151 | 0.181 | 0.333 | **0.609** |
| DCL† | × | × | × | 0.527 | 0.525 | 0.525 | 0.393 | 0.255 | 0.186 | 0.144 | 0.185 | 0.298 | 0.526 |
| | × | ✓ | ✓ | 0.558 | 0.551 | 0.553 | 0.396 | 0.260 | 0.192 | 0.150 | **0.192** | 0.310 | 0.554 |
| | ✓ | × | ✓ | **0.575** | **0.571** | **0.572** | 0.384 | 0.245 | 0.176 | 0.134 | 0.179 | 0.296 | **0.573** |
| | ✓ | ✓ | × | 0.537 | 0.531 | 0.532 | 0.386 | 0.250 | 0.182 | 0.140 | 0.188 | 0.301 | 0.533 |
| | ✓ | ✓ | ✓ | **0.575** | 0.569 | 0.571 | **0.399** | **0.262** | **0.193** | **0.151** | 0.191 | **0.312** | 0.572 |
| PROMPTMRG* | × | × | × | 0.200 | 0.210 | 0.197 | **0.419** | **0.248** | 0.158 | 0.106 | **0.161** | **0.314** | 0.202 |
| | × | ✓ | ✓ | 0.194 | 0.203 | 0.191 | 0.417 | 0.244 | 0.153 | 0.101 | 0.157 | 0.304 | 0.196 |
| | ✓ | × | ✓ | 0.205 | 0.211 | 0.199 | 0.413 | 0.243 | 0.155 | 0.104 | 0.158 | 0.309 | 0.205 |
| | ✓ | ✓ | × | 0.203 | **0.221** | 0.202 | 0.411 | 0.243 | 0.155 | 0.104 | 0.158 | 0.306 | **0.209** |
| | ✓ | ✓ | ✓ | **0.207** | 0.217 | **0.203** | **0.419** | **0.248** | **0.159** | **0.107** | 0.158 | 0.308 | **0.209** |
| CAMANET† | × | × | × | 0.503 | 0.490 | 0.494 | 0.398 | 0.256 | 0.185 | 0.142 | 0.166 | 0.339 | 0.496 |
| | × | ✓ | ✓ | 0.510 | 0.508 | 0.509 | 0.243 | 0.148 | 0.106 | 0.081 | 0.120 | 0.312 | 0.509 |
| | ✓ | × | ✓ | **0.522** | **0.515** | **0.517** | 0.268 | 0.164 | 0.118 | 0.090 | 0.126 | 0.316 | **0.518** |
| | ✓ | ✓ | × | 0.519 | 0.506 | 0.510 | 0.350 | 0.218 | 0.155 | 0.116 | 0.152 | 0.323 | 0.512 |
| | ✓ | ✓ | ✓ | 0.520 | 0.514 | 0.516 | **0.409** | **0.261** | **0.190** | **0.145** | **0.170** | **0.349** | 0.517 |
| DDATR* | × | × | × | **0.308** | 0.282 | 0.257 | 0.428 | 0.253 | 0.160 | 0.107 | 0.161 | 0.314 | 0.282 |
| | × | ✓ | ✓ | 0.305 | 0.277 | 0.251 | 0.425 | 0.249 | 0.157 | 0.105 | 0.161 | 0.309 | 0.278 |
| | ✓ | × | ✓ | 0.305 | 0.293 | 0.249 | 0.432 | **0.258** | **0.165** | **0.111** | **0.164** | **0.321** | 0.282 |
| | ✓ | ✓ | × | 0.301 | 0.304 | **0.265** | 0.430 | 0.253 | 0.159 | 0.106 | 0.163 | 0.312 | **0.290** |
| | ✓ | ✓ | ✓ | 0.281 | **0.308** | 0.259 | **0.433** | 0.257 | 0.164 | 0.110 | 0.163 | 0.313 | 0.283 |
| TGRG† | × | × | × | 0.508 | 0.497 | 0.500 | 0.471 | 0.302 | 0.204 | 0.146 | 0.189 | **0.392** | 0.502 |
| | × | ✓ | ✓ | 0.518 | 0.511 | 0.513 | 0.499 | 0.311 | 0.204 | 0.136 | **0.213** | 0.382 | 0.514 |
| | ✓ | × | ✓ | 0.507 | 0.504 | 0.503 | **0.509** | 0.323 | 0.218 | 0.151 | 0.205 | 0.390 | 0.505 |
| | ✓ | ✓ | × | 0.523 | 0.509 | 0.513 | 0.505 | **0.324** | **0.225** | **0.165** | 0.202 | 0.383 | 0.515 |
| | ✓ | ✓ | ✓ | **0.527** | **0.530** | **0.525** | 0.498 | 0.308 | 0.207 | 0.145 | 0.194 | 0.382 | **0.527** |
| REVTAF* | × | × | × | 0.292 | 0.291 | 0.283 | 0.420 | 0.249 | 0.159 | 0.106 | **0.178** | 0.311 | 0.289 |
| | × | ✓ | ✓ | 0.304 | **0.303** | **0.295** | 0.429 | 0.254 | 0.162 | 0.108 | 0.176 | 0.310 | **0.301** |
| | ✓ | × | ✓ | 0.299 | 0.298 | 0.289 | 0.422 | 0.250 | 0.160 | 0.106 | 0.176 | 0.310 | 0.295 |
| | ✓ | ✓ | × | **0.305** | 0.301 | 0.294 | **0.436** | **0.260** | **0.168** | **0.114** | 0.177 | **0.314** | 0.300 |
| | ✓ | ✓ | ✓ | 0.304 | 0.302 | 0.294 | 0.424 | 0.251 | 0.159 | 0.105 | 0.177 | 0.310 | 0.300 |

**Stage 1: Conflict-Averse Direction Rectification.**    The contribution of conflict rectification is particularly pivotal for the zero-shot models, namely PromptMRG, REVTAF, and DDaTR. Since these models inherit parameters trained on MIMIC-CXR, the auxiliary gradients they generate—derived from multi-label classification or retrieval tasks—are intrinsically biased towards the source domain distribution. When applied directly to IU X-Ray images, these raw gradients often deviate significantly from the optimal generation direction of the target domain. For instance, in PromptMRG, removing Stage 1 leads to a notable regression in the F1-score from 0.203 to 0.191. This empirical evidence suggests that the manifold-guided rectification in Stage 1 acts as a dynamic domain adapter. By projecting the conflicting auxiliary gradients onto the tangent space of the primary objective, Stage 1 effectively filters out source-domain noise while preserving the generalized semantic features. Conversely, for REVTAF, removing Stage 1 marginally outperforms the full model (CE Avg 0.301 vs. 0.300), indicating that forced geometric alignment is counterproductive when cross-domain retrieval gradients degenerate into pure noise. Similarly, its higher NLG scores without Stage 3 confirm that completely discarding these corrupted original gradients prevents flawed retrieval priors from disrupting language fluency.

**Stage 2: Magnitude-Enhanced Energy Injection.**    Stage 2 serves as a critical stabilizer, consistent across both zero-shot and supervised settings. In low-resource scenarios like IU X-Ray, the gradient landscape is often sparse and irregular. The projection operation in Stage 1, while correcting the angular direction, inevitably reduces the gradient norm. Without energy compensation, this reduction can lead to vanishing updates and premature convergence, trapping the model in suboptimal local minima. The results on XProNet demonstrate this effect clearly. Removing Stage 2 leads to a comprehensive degradation in performance, with the average clinical efficacy dropping significantly from 0.609 to 0.579. This confirms that restoring the kinetic energy of the optimization trajectory is indispensable for traversing the rugged loss landscape of small datasets. It ensures that the model retains sufficient momentum to escape saddle points and reach a more robust solution, effectively counteracting the side effects of the geometric projection.

**Stage 3: Adaptive Gradient Fusion.**    For the supervised models trained from scratch, such as WCL, DCL, and TGRG, Stage 3 plays a critical role that extends beyond simple conflict management; it functions as a vital regularization mechanism. Given the severe data scarcity of IU X-Ray, models are highly susceptible to overfitting the training set, often manifesting as high lexical overlap but poor clinical reasoning on unseen data. Stage 3 governs how much the optimizer relies on the structural priors provided by auxiliary tasks versus the data-driven language modeling loss. In the case of WCL, removing Stage 3 causes the F1-score to decline from 0.516 to 0.506. This indicates that a fixed fusion strategy fails to dynamically leverage the global semantic regularization provided by contrastive learning. By adaptively up-weighting the auxiliary gradients when the synergy is high—consistent with the higher fusion coefficients observed in our hyperparameter analysis—Stage 3 prevents the generator from collapsing into memorization. It forces the model to respect the auxiliary structural constraints, ensuring that the learned representations remain robust and generalizable to the test set.

**Trade-off Analysis.**    Finally, we observe an interesting trade-off regarding Natural Language Generation metrics in specific baselines. For models like XProNet, the variant without Stage 1 occasionally yields higher BLEU scores than the full model. However, this increase in surface-level fluency comes at the cost of clinical accuracy, as evidenced by the lower Precision and Recall scores. This phenomenon suggests that without the constraint of conflict rectification, the model may overfit to frequent linguistic patterns in the training captions, generating fluent but factually incorrect reports. CAME-Grad prioritizes clinical correctness by suppressing these ungrounded language priors, ensuring that the generated reports are not only readable but, more importantly, diagnostically accurate.

## F. Performance of Auxiliary Classification Task

Assessing the auxiliary classification task is fundamental to ensuring that gradient rectification does not compromise the underlying clinical supervisory signals. In the multi-task radiology report generation (RRG) framework, the classification branch provides critical diagnostic anchors that ground the primary generative manifold.

To comprehensively evaluate this multi-label classification performance, we report the Area Under the Curve (AUC), Precision, Recall, and F1-score. Given the inherent long-tailed disease distribution in the MIMIC-CXR dataset, we calculate both Macro-averaged metrics (which treat each of the 14 observation classes equally) and Micro-averaged metrics (which aggregate contributions globally to account for class imbalance). This dual perspective ensures a rigorous and unbiased assessment of the auxiliary branch's diagnostic capacity.

As summarized in Table F.1, CAME-Grad demonstrates robust performance preservation and enhancement on the auxiliary

branch across different backbones. On the PromptMRG baseline, the Macro F1-score improves from 0.404 to 0.415, while the Micro F1-score increases from 0.585 to 0.593. For the REVTAF backbone, CAME-Grad yields a notable gain in the Macro F1-score from 0.595 to 0.610. Although the Macro Precision of REVTAF exhibits a minor reduction from 0.654 to 0.651, the overall diagnostic efficacy reflected by the AUC and F1-score consistently improves. These findings confirm that CAME-Grad effectively resolves mutual interference and achieves synergistic optimization across both tasks, allowing the model to reach a superior equilibrium between diagnostic accuracy and report quality.

*Table F.1.* Classification performance comparison on the MIMIC-CXR dataset. The best results for each baseline pair are highlighted in bold. AUC and F1-score provide a comprehensive evaluation of the auxiliary diagnostic branch.

| MODEL | PUBLICATION | MACRO METRICS | | | | MICRO METRICS | | | |
|---|---|---|---|---|---|---|---|---|---|
| | | AUC ↑ | PREC. ↑ | REC. ↑ | F1 ↑ | AUC ↑ | PREC. ↑ | REC. ↑ | F1 ↑ |
| PROMPTMRG | AAAI'24 | 0.811 | 0.439 | 0.387 | 0.404 | 0.874 | 0.610 | 0.562 | 0.585 |
| + CAME-GRAD | | **0.814** | **0.460** | **0.397** | **0.415** | **0.877** | **0.617** | **0.571** | **0.593** |
| REVTAF | ICCV'25 | 0.905 | **0.654** | 0.562 | 0.595 | 0.936 | 0.741 | 0.702 | 0.721 |
| + CAME-GRAD | | **0.907** | 0.651 | **0.588** | **0.610** | **0.938** | **0.742** | **0.721** | **0.732** |

## G. Extended Qualitative Analysis

In this section, we provide an extended qualitative evaluation to intuitively demonstrate the clinical accuracy and coherence of the reports generated by CAME-Grad. Figure G.1 compares the qualitative results of the baseline PromptMRG and the CAME-Grad equipped PromptMRG on the MIMIC-CXR test set.

As illustrated in the case, the Ground Truth (GT) clearly indicates "areas of moderate retrocardiac atelectasis" along with the presence of multiple support devices. However, the baseline PromptMRG exhibits serious clinical limitations driven by visual tunnel vision. It allocates excessive attention to the support devices in the upper mediastinum, generating redundant descriptions of catheter positions (e.g., "projects 4 cm above the carina"). Consequently, it completely fails to attend to the retrocardiac region—indicated by the cold blue regions in its attention map—and misses the critical finding of atelectasis.

In contrast, the report generated by the model optimized by CAME-Grad aligns highly with the gold standard. It not only accurately verifies the support devices but also successfully penetrates the anatomical occlusion to diagnose "subsequent areas of atelectasis". The attention map further confirms that the model successfully attended to the retrocardiac region highlighted in cyan/yellow, a critical pathological area missed by the baseline model. This demonstrates that CAME-Grad successfully guides the model to explore distinct discriminative features beyond salient objects (like tubes) by rectifying gradient direction and enhancing exploration momentum. Consequently, it significantly reduces the risk of clinical missed diagnoses and greatly improves the diagnostic credibility of the reports.

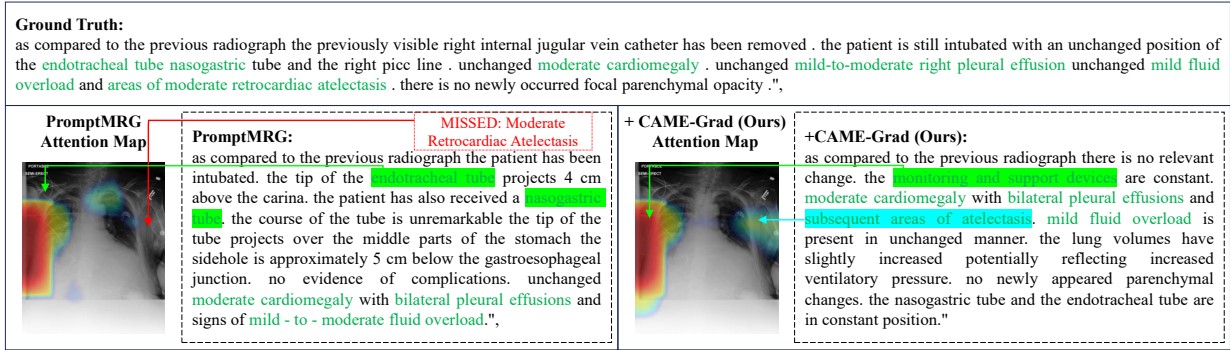

*Figure G.1.* Qualitative comparison and attention visualization on the MIMIC-CXR test set. The Ground Truth report is shown at the top. The PromptMRG baseline (left) is distracted by the support devices in the upper mediastinum and misses the "moderate retrocardiac atelectasis" (marked in red). In contrast, CAME-Grad (right) successfully allocates attention to the retrocardiac region (highlighted by the cyan arrow and heatmap hotspot) and correctly identifies the atelectasis, demonstrating superior anatomical consistency.

