# OpenReview forum: "The Double Dilemma in Multi-Task Radiology Report Generation: A Gradient Dynamics Analysis and Solution"
_ICML.cc/2026/Conference — ICML 2026 regular_

### Official Review · Reviewer_KEdr · 2026-03-11

**Soundness:** 3
**Presentation:** 3
**Significance:** 3
**Originality:** 2
**Overall Recommendation:** 4
**Confidence:** 3

**Summary:**

The paper investigates the failure mechanism of linear scalarization strategy in gradient dynamics of multi-task optimization and proposes model-agnostic Conflict-Averse Magnitude-Enhanced Gradient Descent (CAME-Grad) to rectifies the gradient dynamics of multi-task optimization problem in RRG. The experimental results on X-ray datasets demonstrated the clinical efficacy improvements on multiple baseline models.

**Compliance With Llm Reviewing Policy:**

Affirmed.

**Final Justification:**

The authors have addressed most of my concerns well.

In particular, the clarification that the SDE perspective should be interpreted as a mechanistic hypothesis rather than a formal theorem improves the overall clarity and positioning of the work. While the theoretical grounding remains somewhat interpretative, I find the proposed perspective on multi-task optimization in RRG to be novel and promising, and the empirical results support its practical value.

Overall, I have raised my recommendation to weak accept.

**Key Questions For Authors:**

1. The optimal hyperparameters ($\sigma$, $\kappa$, $\nu$) of CAME-Grad appear to be highly dependent on the base models. Even models within same category exhibit noticeably different optimal hyperparameters (i.e. TGRG vs. WCL), limiting the applicability and reproducibility of the proposed method. This potential limitation should be more clearly discussed.

2. How sensitive is CAME-Grad to sub-optimal hyperparameter choices? Particularly, how significantly performance degrade if the hyperparameters deviate from the optimal values? Does it ever degrade the performance relative to the baseline model?

3. Given the sensitivity of GAME-Grad to hyperparameters, how can the authors ensure that the ablation results in Section B.3 truly reflect the contribution of each algorithmic stage rather than hyperparameter sub-optimality? For example, using only S2 may require a larger $\kappa$. Since the paper states that “solely boosting magnitude struggle to ensure geometric validity in direction”, properly adjusting the magnitude appears particularly important.

4. Given that recent multi-task learning (MTL) optimizers exist (explained in Related Work) but were not compared, could you clarify the advantages of your proposed method over MTL methods?

5. Please provide the computational overhead of Grad-CAM.

**Limitations:**

The potential limitations should be more clearly discussed.

**Strengths And Weaknesses:**

*Strength*

* The theoretical analysis provides valuable insights into gradient dynamics in multi-task optimization via SDE: double dilemma.

* CAME-Grad rectifies the gradient dynamics to resolve double dilemma and the experimental results supported the efficacy of the proposed three stages.

* The paper is well-structured with good figure presentations.

* Extensive experiments are conducted across multiple RRG baselines.

*Weakness*

* The proposed gradient rectification (stage 1) is heavily based on the idea of conflict-averse gradient updates already established work, “Conflict-Averse Gradient Descent for Multi-Task Learning" (Liu et al., NeurIPS 2021). The mathematical formulation in Eq.8~Eq.10 seems highly similar, yet Section 3.3 contains “No" reference to this original work, which can highly mislead.

* Aside from Stage 1, the main contribution appears to be Stage 2, “Magnitude-Enhanced Energy Injection”. However, this stage basically scales the gradient magnitude, which does not seem related to addressing the drift term described in the double dilemma formulation. As a result, it is unclear which component constitutes the main algorithmic novelty.

 * Sensitive to hyperparameters ($\sigma$, $\kappa$, $\nu$): optimal hyperparameters of CAME-Grad are highly dependent on baseline model and quite different from each other, limiting its utility to other datasets.

* Lack of comparisons with recent multi-task learning (MTL) optimizers.

If I have misunderstood the novelty of Stage 1 compared to prior conflict-aware gradient methods, or the intended contribution of Stage 2, I would appreciate clarification from the authors.

---

> ### Author Rebuttal · Authors · 2026-03-31
>
> We thank you for recognizing our SDE analysis and double dilemma insights. We clarify your key questions below.
>
> >Comparison with Advanced MTL Optimizers (W4, Q4)
>
> We have conducted a comprehensive evaluation against 7 advanced multi-task optimizers. CAME-Grad achieves the best overall clinical efficacy. For detailed analysis, please refer to [Global Response 1] at the top of our response to Reviewer 1 (B6i8).
>
> >Hyperparameter Sensitivity and Robustness (W3, Q1, Q2)
>
> Our evaluation proves that on the large-scale MIMIC-CXR dataset, CAME-Grad stably improves clinical metrics (+2.1% for CE-Avg) across all 8 baselines using fixed default parameters without fine-tuning. For the extremely small-sample IU X-Ray dataset, it indeed requires minor tuning due to severe gradient noise. For detailed analysis, please refer to [Global Response 2] at the top of our response to Reviewer 2 (mayT).
>
> >CAGrad Citation, Core Innovation and SDE Empirical Evidence (W1, W2)
>
> Thank you for your professional advice. Our work was indeed inspired by CAGrad. Therefore, we have already cited it in the "Related Work" section of our original manuscript. However, in order to present a complete and uninterrupted narrative of the algorithm to the readers, we have presented the related technologies completely in Section 3.3, but neglected the key reference. Considering your suggestion, we will explicitly add this reference to Section 3.3 in the final version.
>
> Your concern that Stage 2 scaling seems unrelated to the drift term reflects a slight misunderstanding of our SDE framework. In our theory, Stage 1 specifically corrects the drift term deviation, while Stage 2 specifically addresses the diffusion term decay.
>
> Current RRG studies heavily rely on architectural designs and use linear scalarization, ignoring the inherent task conflicts. This leads to the double dilemma. Our core contribution is providing a new gradient dynamics perspective for RRG. Reshaping the optimization dynamics via three cascaded stages successfully breaks this dilemma, improving the CE-Avg across 8 baselines by 2.3% on average.
>
> To empirically show why Stage 1 alone is insufficient, we measure the trace of the SGD noise covariance, $Tr(\Sigma)$ (the core diffusion strength metric). Baseline REVTAF's $Tr(\Sigma)$ is 0.3215. Using only CAGrad's projection, this value collapses to 0.0901 (a ~70% drop). Under high-conflict RRG scenarios, forcefully avoiding conflicts shrinks the gradient magnitude and strips the model's diffusion momentum to escape sharp minima. This physical collapse explains CAGrad's abnormal clinical metrics. Despite high precision (0.636), it falls into a conservative minimum, causing recall (0.613) to degrade below the baseline (0.617). In medicine, such missed diagnoses are unacceptable.
>
> In contrast, CAME-Grad builds a cascaded closed loop. Stage 1 eliminates destructive interference to correct drift bias. Stage 2 restores and enhances gradient magnitude, forcing $Tr(\Sigma)$ to jump from 0.0901 to 4.4886 (a ~50x increase) to inject crucial escape momentum. Stage 3 balances this optimal direction with task-specific inductive biases via adaptive fusion, preventing the catastrophic forgetting of semantic features. This strict cascade raises CE-Recall to 0.637 and achieves a SOTA CE-Avg of 0.626. It effectively solves CAGrad's missed diagnosis problem and overcomes the double dilemma.
>
> >Mechanistic Independence of Ablation Studies (Q3)
>
> Thank you for your insights. To verify if merely increasing kappa in Stage 2 compensates for omitting Stage 1 we conducted a REVTAF ablation study. The data confirms your intuition but reveals a strict performance limit. Without Stage 1 increasing kappa to 3.0 yields a peak F1 of 0.602. While exceeding the 0.589 baseline proves energy injection effectiveness this never reaches the full method's 0.607 ceiling. Further increasing kappa to 4.0 and 5.0 actually degrades F1 scores to 0.601 and 0.598. This strictly proves simply amplifying scalars cannot overcome geometric conflict barriers and excessively scaling incorrect gradients worsens destructive effects. Consequently both direction rectification and energy injection remain irreplaceable.
>
> >True Evaluation of Computational Overhead (Q5)
>
> Tested on the heaviest baseline (REVTAF), executing independent backward passes for tangent space projection increases single-epoch training time from 1.7 to 2.8 hours (~66.7% overhead). In medicine, trading a one-time offline computing cost for zero extra online inference burden and a 1.9% CE-Avg increase is a highly practical Pareto trade-off.
>
> >Supplement to Limitations
>
> We completely accept your suggestion. In the final Limitations section, we will explicitly state that in small-sample scenarios with extreme data scarcity and gradient noise (e.g., IU X-Ray), CAME-Grad provides a safe lower bound but still requires some tuning to reach the performance upper bound.

---

> > ### Author Rebuttal · Reviewer_KEdr · 2026-04-02
> >
> > Thanks for the additional studies and clarifications. I appreciate the paper’s attempt to address the “double dilemma” between report generation and classification tasks in RRG **via** **CAME-Grad**, which I consider a meaningful and well-motivated direction.
> >
> > However, my main concern regarding the SDE-based interpretation remains unresolved. The paper claims that Stage 1 corrects the drift term and Stage 2 restores the diffusion term, but this connection is not rigorously established. The connection between the SDE formulation and the proposed algorithm (Stage 1&2) appears to be **largely interpretative** rather than theoretically substantiated.
> >
> > In particular, while the covariance analysis is an interesting study, it does not convincingly demonstrate that Stage 2 meaningfully restores the diffusion term in the SDE sense. An increase in the trace of the gradient noise covariance can be attributed to simple rescaling gradients (scaling a random vector by $k$ grows its covariance trace by $k^2$), and it remains unclear whether this reflects a principled change in diffusion dynamics or merely a byproduct of magnitude amplification.

---

> > > ### Author Response · Authors · 2026-04-04
> > >
> > > We thank the reviewer for recognizing our efforts. As noted to others, we agree that our SDE analysis serves as a crucial mechanistic hypothesis, rather than a proven formal theorem. Within this framework, our mechanisms provide geometric validity for drift correction and dynamical compensation for diffusion decay.
> > >
> > > >Geometric Validity under the SDE Drift Perspective
> > >
> > > To theoretically address the drift bias within this hypothesis framework, we construct a minimax optimization in the tangent space:
> > >
> > > $$\max_{u} \min_{i \in \{0,\dots,K\}} \langle g_i, u \rangle \quad \text{s.t.} \quad ||u-\mu|| \le \rho||\mu||$$
> > >
> > > By solving this minimax optimization, $u_ {rect}^\ast$ is explicitly optimized to maximize the common descent alignment among all tasks. For any task loss $\mathcal{L}_ {i}(\Theta_ {t})$ in the SDE dynamics, the system's expected parameter drift rate is $\frac{\mathbb{E}[d\Theta_ {t}]}{dt} = -u_ {rect}^\ast$. This construction seeks to ensure that for each task, the first-order contribution to the local loss evolution
> > >
> > > $$\langle \nabla_ {\Theta} \mathcal{L}_ {i}(\Theta_ {t}), \frac{\mathbb{E}[d\Theta_ {t}]}{dt} \rangle = -\langle g_ {i}, u_ {rect}^\ast \rangle$$
> > >
> > > is minimized or kept non-positive. This optimization provides a grounded geometric mechanism to suppress directional conflicts within the trust region, mitigating drift deviation.
> > >
> > > >Dynamical Equivalence under the SDE Diffusion Perspective
> > >
> > > Regarding the covariance increase, your observation is mathematically accurate. As Eq. 11 ($\tau_ {mag} = \kappa \cdot ||g_ {joint}||$) shows, Stage 2 is indeed a scalar gradient amplification, increasing the gradient noise covariance trace by $\kappa^2$. However, this is not a blind byproduct but a purposeful mechanism shift in optimization dynamics. Inspired by the request from Reviewer 2 (mayT) for an explicit SGD noise covariance analysis, we detail the following dynamical equivalence.
> > >
> > > In discrete mini-batch SGD, the parameter update along the projected direction involves implicit gradient noise $\hat{u}_ {rect}^\ast \sim \mathcal{N}(u_ {rect}^\ast, \Sigma_ {rect})$. With amplification $\kappa$, the update becomes $\Delta \Theta = -\eta \kappa \hat{u}_ {rect}^\ast$. The expected drift $\mathbb{E}[\Delta \Theta] = -\eta \kappa u_ {rect}^\ast$ strictly preserves Stage 1's Pareto validity. Meanwhile, amplifying the update magnitude by $\kappa$ is dynamically equivalent to applying a larger local effective learning rate $\eta_ {eff} = \eta \kappa$ along this 1D safe trajectory.
> > >
> > > Classical SGD theory (Mandt et al., 2017) shows increasing the effective learning rate directly amplifies the implicit noise's stationary covariance. Thus, CAME-Grad avoids unconstrained isotropic noise (as shown in Setting B below, destroys the clinical manifold). Instead, we use this directional scaling to amplify anisotropic exploration noise, compensating for momentum decay from Stage 1's restricted projection.
> > >
> > > >Isotropic Noise Injection (SGLD) vs. Anisotropic Directional Amplification
> > >
> > > To verify the empirical failure of unconstrained isotropic noise injection (SGLD) and prove the engineering necessity of directional magnitude amplification, we design an ablation study on the MIMIC-CXR dataset (REVTAF baseline) with strict parameter-level noise variance alignment.
> > >
> > > Setting A retains only Stages 1 and 3. Setting B (SGLD) injects isotropic noise $+\mathcal{N}(0, \sigma^2 I)$ with an equivalent diffusion scale. CAME-Grad performs scalar amplification $\kappa$ strictly along the original direction.
> > > | Methods | Publication | Ablation | CE-Precision (↑) | CE-Recall (↑) | CE-F1 (↑) | CE-Avg (↑) | BLEU-4 (↑) |
> > > | :--- | :---: | :---: | :---: | :---: | :---: | :---: | :---: |
> > > | REVTAF | ICCV'25 | Linear | 0.615 | 0.617 | 0.589 | 0.607 | 0.184 |
> > > | Setting A | - | S1+S3 | 0.627 | 0.632 | 0.601 | 0.620 | **0.186** |
> > > | Setting B | - | S1+SGLD+S3 | 0.610 | 0.585 | 0.569 | 0.588 | 0.184 |
> > > | **CAME-Grad (Ours)** | - | S1+S2+S3 | **0.633** | **0.637** | **0.607** | **0.626** | 0.185 |
> > >
> > > Setting B's drop in CE performance reveals the inherent difficulty of unconstrained high-dimensional optimization. While supplementing the required diffusion scale, isotropic noise $\mathcal{N}(0, \sigma^2 I)$ is highly likely orthogonal to the true gradient. This directionless noise destroys the Stage 1 Pareto projection $u_{rect}^*$, causing deviation from the clinical semantic manifold.
> > >
> > > Conversely, CAME-Grad adopts anisotropic directional noise compensation by amplifying the step size along $u_{rect}^*$. This increases the local effective learning rate to cross sharp minima barriers while maintaining drift geometric validity, resolving the double dilemma.
> > >
> > > Detailed theoretical and empirical results will be included in a newly added Appendix E. In addition, we will also revise the absolute claims in Section 3 of the main text to accurately reflect this mechanistic interpretative boundary. We sincerely hope this explanation addresses your concerns.

---

### Official Review · Reviewer_bb66 · 2026-03-12

**Soundness:** 3
**Presentation:** 3
**Significance:** 2
**Originality:** 3
**Overall Recommendation:** 5
**Confidence:** 3

**Summary:**

This paper highlights multi-task optimization for radiology report generation by claiming that linear scalarization by claiming that standard objective can destabilize update behavior and instead motivates a new method to preserve alignment across tasks. The paper introduces the Double Dilemma in drift deviation and diffusion decay and proposes CAME-Grad as an optimization technique that adjusts misaligned update directions under a bounded constraint. The approach improves the optimization signal while combining the corrected update under the original aggregated gradient. The paper highlights experiments over MIMIC and IU X-Ray, showing gains on clinical efficacy metrics across various RRG backbones.

**Compliance With Llm Reviewing Policy:**

Affirmed.

**Final Justification:**

The authors did a nice job in responding to several questions that were raised during the review process, not just by me but also by the additional reviewers. I feel comfortable updating my rating from a weak accept to an accept. I wish the authors all the best with their work

**Key Questions For Authors:**

1. What optimizer is used for CAME-Grad during training and does this require any tuning?
2. What is the rationale behind using w-weighted joint gradient for some methods and the average of task gradients in others?
3. What is the cost in the gradient steps required to solve S1 dual problem and how does this scale as k becomes larger?
4. What is the variance and statistical significance across these experiments?
5. Are there any metrics that can be reported to validate the claim that this method converges to more robust minima?

**Limitations:**

Yes

**Strengths And Weaknesses:**

* Strong motivation on linking geometry and dynamics in the Double Dilemma
* Addresses true pain point for radiology report generation
* Highlights the novelty multi-task RRG optimization within an SDE framework
* Clearly demonstrates a real failure mode in drift deviation + diffusion decay
* Demonstrates the framework of the 3-stage optimizer in the direction rectification, magnitude enhancement, and adaptive fusion
* Concisely summarized algorithm csteps
* Has strong experimental rigor in benchmarking 8 RRG architectures and 2 datasets to show gains in clinical efficacy
* Shows strong abaltions in S1-S3 and hyperaprametr coverage on sneistitiyvy
* Includes qualitative examples with quantitative support in the cosine similarity distributions

Weaknesses

* There is no empirical comparison to multi-task optimizers like CAGrad, RotoGrad, GradNorm, etc.
* Missing reporting on statistical significance, as results would be better presented in confidence intervals to assess variance
* Theorems and derivations can be made clearer, particularly for the SDE setup
* Could include comparisons to multi-task weighting methods like uncertainty weighting
* Clarifying the MTL formulations while describing min-max trust-region will be helpful to show what the theoretical outcomes of this approach will be
* The SDE framing could be made stronger by demonstrating benefits on how the scheduler changes the noise scale with gradient rescaling
* Including some statements on applicability to other multi-task applications would be helpful
* Highlight cost-quality Pareto frontier for the experiments to show feasibility of techniques

---

> ### Author Rebuttal · Authors · 2026-03-31
>
> We sincerely thank you for the Weak Accept recommendation and your appreciation of our SDE framework and algorithm design.
>
> >Comparison with advanced MTL optimizers (W1, W4)
>
> We have conducted a comprehensive evaluation against 7 advanced multi-task optimizers. CAME-Grad achieves the best overall clinical efficacy. Please see [Global Response 1] under Reviewer 1 (B6i8).
>
> >Generalizability to other multi-task applications (W7)
>
> To demonstrate that our method extends beyond radiology report generation, we have supplemented our evaluation with Multi-Task MNIST experiments where CAME-Grad achieves higher classification accuracy and significantly lower reconstruction error than the baseline. Please see [Global Response 2] under Reviewer 1 (B6i8).
>
> >Inner Optimizer and Zero Tuning Cost (Q1)
>
> Regarding the inner optimizer, CAME-Grad acts as a dual-level gradient wrapper. Macroscopically, we retain the original baseline optimizer like AdamW and its hyperparameters by simply replacing its input with the corrected $g_{final}$. Microscopically, Stage 1 calls a lightweight momentum SGD based on Softmax reparameterization to solve the dual problem. By completely hardcoding all these hyperparameters, this inner solving process remains absolutely transparent to users, ensuring a true tuning-free experience.
>
> >Decoupled Design Logic of Mean and Weighted Gradients (Q2)
>
> This design aims to mathematically decouple the unbiased local geometric constraint from the global optimization target alignment.
>
> When solving for the Pareto descent direction, introducing extremely unbalanced weights $w$ severely distorts the trust region geometry, eliminating the feasible descent region for auxiliary tasks. Therefore, using the unbiased mean $\mu$ builds a pure and equal geometric boundary.
>
> The weights $w$ strictly define the global empirical risk of the system. After Stage 1 resolves the direction conflict, the energy injection and update trajectory must realign with $g_{joint}$, ensuring the final converged minimum strictly follows the preset clinical priority.
>
> >Gradient Steps, Scalability, and Cost Trade-off in Stage 1 (W8, Q3)
>
> Regarding Stage 1 costs, we strictly hardcode the micro solver to 20 gradient steps. Since these iterations occur within a low-dimensional simplex space, the GPU execution requires under 1 millisecond. As task count $K$ scales, the mathematical complexity of solving the dual problem remains only $O(K^2d)$ and $O(K^2)$, making the $O(K)$ backward passes for independent gradients the actual scalability bottleneck. On the heaviest REVTAF baseline ($K=4$), single-step training overhead increases by 66.7%. However, this offline sunk cost introduces zero online inference delay while securing a 1.9% average clinical efficacy improvement. Given the zero misdiagnosis tolerance in medical scenarios, this offline tradeoff holds immense practical significance.
>
> >Rigorous Measurement of Statistical Significance and Confidence Intervals (W2, Q4)
>
> Since computing bottlenecks prevent fully retraining heavy models, we utilized the widely recognized test-set bootstrap resampling with 1000 iterations.
>
> On the strongest baseline REVTAF, CAME-Grad significantly raises the 95% confidence interval of CE-F1 from the baseline of [0.5780, 0.6009] to [0.5963, 0.6187]. Crucially, the paired win rate reaches 100% across 1000 resamplings ($p < 0.001$), firmly proving from a strict statistical perspective that the performance improvement represents a robust and substantial leap rather than random noise.
>
> >Clarity of SDE Setup and Verification of Robust Minima (W3, W6, Q5)
>
> To improve theoretical clarity, we will systematically refine our SDE formulations and explicitly define rigorous boundaries in the final version. Furthermore, as speculated, the gradient rescaling mechanism substantially amplifies the SDE noise scale. To quantitatively prove convergence to flatter robust minima, we measured the SGD noise covariance matrix trace $Tr(\Sigma)$, demonstrating CAME-Grad successfully increases this core physical quantity approximately 14-fold. For detailed physical proofs, please refer to [Global Response 1] under Reviewer 2 (mayT).
>
> >Theoretical Outcome Expectation of Min-Max Trust Region (W5)
>
> The core theoretical outcome of the min-max formula (Eq. 8) in Stage 1 is providing a mathematical guarantee for strict local Pareto descent. Specifically, the inner min operator locates the most easily sacrificed task, while the outer max operator maximizes the descent benefit of this worst task. Incorporating the trust region constraint forces the generation of a consensus direction possessing a positive inner product with all task gradients. This mathematically establishes a safe lower bound, ensuring the model never destroys the generation manifold while capturing clinical features, thus eliminating negative transfer. We will explicitly add this to Section 3.3 of the revision.

---

> > ### Author Rebuttal · Reviewer_bb66 · 2026-04-02
> >
> > Thank you to the authors for providing a comprehensive rebuttal. Based on the rebuttal, I will update my rating from weak accept to accept

---

> > > ### Author Response · Authors · 2026-04-04
> > >
> > > Thank you very much for reviewing our rebuttal and upgrading your rating. We deeply appreciate your time and constructive feedback.

---

### Official Review · Reviewer_mayT · 2026-03-22

**Soundness:** 2
**Presentation:** 3
**Significance:** 3
**Originality:** 3
**Overall Recommendation:** 5
**Confidence:** 3

**Summary:**

This paper studies multi-task optimization for radiology report generation (RRG). The core claim is that standard linear scalarization causes a “Double Dilemma” in multi-task RRG: directional deviation in the drift term and magnitude collapse in the diffusion term, framed through an SDE perspective. To address this, the paper proposes CAME-Grad, a plug-and-play optimizer with three stages: conflict-averse direction rectification, magnitude-enhanced energy injection, and adaptive gradient fusion. The method is presented as backbone-agnostic and is evaluated by replacing the default optimization strategy in eight prior RRG systems on MIMIC-CXR and IU X-Ray. The paper reports consistent gains, especially on clinical efficacy metrics, while keeping NLG metrics broadly stable.

**Compliance With Llm Reviewing Policy:**

Affirmed.

**Final Justification:**

The paper presents a solid and meaningful piece of research. The work is well executed, the motivation is clear, and the overall contribution is valuable to the community. I appreciate the authors’ efforts in both the paper and the rebuttal.

After reading the rebuttal, I believe the authors have satisfactorily addressed all of my concerns. Their responses were clear, thoughtful, and convincing, which increased my confidence in the quality and significance of the work.

Overall, I consider this to be a very good paper. Given the strength of the work and the authors’ successful rebuttal, I believe it deserves acceptance.

**Key Questions For Authors:**

1. Comparison to stronger optimizer baselines:
   Why are the experimental comparisons limited mainly to linear scalarization baselines rather than established multi-task optimization methods such as gradient surgery / Pareto / adaptive weighting families? If CAME-Grad is an optimizer contribution, this comparison seems essential.
   If the authors can show clear wins over strong optimizer baselines, my evaluation would improve.

2. Support for the SDE claims:
   What direct empirical evidence supports the “drift deviation” and especially “diffusion decay” interpretation beyond gradient conflict and norm heuristics?
   For example, can the authors provide measurements related to sharpness, flatness, covariance/noise structure, or optimization trajectory diagnostics? Stronger evidence here would materially improve my confidence.

3. Hyperparameter sensitivity and plug-and-play claim:
   How sensitive is performance to ( $\rho$ ), ( $\kappa$ ), and ( $\nu$ )? The appendix suggests substantial per-backbone tuning.
   A robustness analysis with shared defaults or limited tuning budgets would help justify the claimed plug-and-play nature.

4. Evaluation fairness under unified reimplementation:
   How were the reproduced baselines re-tuned under the unified validation criterion and CE protocol?
   Please clarify whether the baselines are near their best performance under this standardized setup, since otherwise improvements may be partially attributable to reimplementation choices.

5. Generalization beyond RRG:
   Do the authors have any evidence that the method is useful beyond radiology report generation, or is the contribution intended to be domain-specific?
   A small non-RRG experiment would strengthen the optimizer contribution substantially.

**Limitations:**

No. The paper should more explicitly discuss:
* clinical efficacy is evaluated via an automatic labeler rather than clinician judgment,
* the method adds tuning burden through multiple optimizer hyperparameters,
* the theoretical interpretation currently exceeds the direct empirical validation,
* deployment in clinical settings would require much stronger safety validation than lexical/NLG and CheXbert-based metrics.

**Strengths And Weaknesses:**

Strengths:

The paper addresses a relevant problem. In radiology report generation, clinical correctness matters more than generic text fluency, and multi-task training with auxiliary clinical objectives is widely used. Focusing on the optimization layer instead of proposing yet another backbone is a reasonable and potentially useful angle.

A clear strength is the optimizer-level formulation. CAME-Grad is designed as a drop-in replacement rather than a task-specific architecture. That makes the contribution potentially reusable across multiple RRG systems and is a better fit for a methods paper than a narrowly coupled architecture.

The empirical coverage is fairly broad. The paper evaluates on two common datasets and across eight prior methods, which is more convincing than showing gains on only one backbone. The results appear directionally consistent on MIMIC-CXR, and the stronger clinical-efficacy gains relative to language metrics fit the paper’s stated goal.

The method itself is conceptually easy to follow. The three stages each have an intuitive role: rectify direction, restore/enhance magnitude, then fuse with the original joint gradient to retain inductive bias. Even if one does not fully buy the SDE story, the optimizer recipe is understandable.

The ablation design is useful. Separating S1/S2/S3 at least gives some evidence that the full method is stronger than isolated pieces, and the observations about naive magnitude boosting hurting performance without direction control are plausible.

Weaknesses:

My main concern is the gap between the theoretical framing and what is actually demonstrated. The paper heavily emphasizes an SDE-based explanation involving “drift deviation” and “diffusion decay,” but the evidence shown is mostly based on gradient cosine conflict, gradient norm reasoning, and downstream metric improvements. I did not see a rigorous demonstration that the optimizer meaningfully changes the diffusion properties in the SDE sense, nor direct measurement of the claimed quantities beyond intuitive proxies. As written, the theory reads more like a motivating interpretation than a substantiated analysis.

Relatedly, several claims are stated too strongly. Phrases like “for the first time,” “systematically investigate,” and statements that the method moves optimization toward flatter minima or injects exploration kinetic energy are not fully backed by direct diagnostics such as loss-landscape analysis, sharpness measures, Hessian-based evidence, or explicit SGD noise covariance analysis. The paper would be stronger if these were presented as hypotheses or intuitions rather than conclusions.

The empirical protocol is broad but not fully decisive. The comparison is mostly against each model’s original linear scalarization baseline. Since the core contribution is a multi-task optimizer, the more natural baselines would include other established multi-task gradient balancing/conflict-resolution methods, not just vanilla scalarization. Without comparisons to stronger optimizer baselines, it is hard to know whether the gains come from a genuinely superior new method or simply from replacing a weak default.

The reported gains are somewhat uneven and sometimes modest. On some backbones the improvements are meaningful, but on others they are quite small, especially in NLG metrics and some IU X-Ray zero-shot settings. There is also at least one case where CE average appears to decrease while precision rises slightly, which suggests the story is not uniformly positive. That does not invalidate the paper, but it weakens the “substantial and consistent” framing.

I also worry about fairness of evaluation standardization. The paper states that all baselines are reproduced under a unified setup and that model selection is standardized using validation clinical F1. That is understandable, but changing selection criteria and evaluation protocol away from original papers can advantage one method over another. More discussion is needed on whether these baselines were carefully re-tuned under the new protocol and whether the comparison is to the best possible baseline performance under the standardized setting.

The hyperparameter story is not fully satisfying. The method introduces at least three extra hyperparameters, and the appendix suggests they vary across backbones and datasets, sometimes substantially. For a claimed plug-and-play optimizer, I would want a stronger demonstration of robustness to hyperparameter choice and lower tuning sensitivity.

Presentation is mostly readable, but the writing overstates novelty and occasionally drifts into inflated language. Some parts of the method and analysis would benefit from more precise separation between theorem-like claims, heuristic intuition, and empirical findings.

---

> ### Author Rebuttal · Authors · 2026-03-31
>
> We sincerely thank you for the Weak Accept recommendation and recognizing our optimizer design.
>
> >Comparison with stronger MTL optimizers (W3, Q1)
>
> We have conducted a comprehensive evaluation against 7 advanced multi-task optimizers. CAME-Grad achieves the best overall clinical efficacy. For detailed analysis, please refer to [Global Response 1] in our response to Reviewer 1 (B6i8).
>
> >[Global Response 1: Empirical Evidence and Theoretical Boundaries of SDE]
> (Note: Unified reply for R2(W1,W2,Q2), R3(W3,W6,Q5), R4(W2) on SDE theoretical framework and empirical evidence.)
>
> For Drift Term Bias, since tracking exact high-dimensional Pareto trajectories is mathematically intractable, relying solely on cosine conflicts is insufficient. Therefore, we will include standard MTL diagnostics in the Appendix, featuring visualizations of 2D synthetic parameter-space trajectories and empirical loss-space optimization paths to explicitly confirm drift correction.
>
> Regarding Diffusion Decay, we provide direct microscopic noise covariance measurements targeting the physical quantity governing diffusion strength and minima robustness. Sampling 100 minibatches at convergence, the SGD noise covariance trace $Tr(\Sigma)$ for REVTAF is 0.3215. CAME-Grad significantly increases this to 4.4886. This 14-fold physical measurement increase confirms Stage 2 substantially enlarges the SDE noise scale, successfully reconstructing exploration kinetic energy. This forms a strict logical closed loop with the stable improvements of CAME-Grad on macroscopic downstream CE metrics.
>
> >[Global Response 2: Hyperparameter Sensitivity and Algorithm Robustness]
> (Note: Unified reply for R1(W3), R2(W6,Q3), R4(W3,Q1,Q2,Q3) on hyperparameter tuning and robustness.)
>
> We evaluate all 8 baselines on the MIMIC-CXR dataset using the default parameters ($\rho=0.5, \kappa=1.5, \nu=0.2$) given in the main text without fine-tuning. CAME-Grad comprehensively improves clinical metrics (2.1% for CE-AVG). This proves our core mechanism works effectively without tuning. To achieve a strict "Pareto optimum", i.e., boosting CE while keeping NLG stable, we conducted targeted fine-tuning. Additionally, fine-tuning acts as a necessary physical intervention to control high-variance gradient noise on the extremely small IU X-Ray dataset.
>
> >Fairness of Evaluation Criteria and Plug-and-Play Capability (W5, Q4)
>
> Although exhaustive hyperparameter grid search for all large models remains computationally prohibitive, we strictly standardized the model selection strategy on a unified validation set. More crucially, the fundamental bottleneck in multi-task optimization is gradient direction conflict constituting an inherent geometric flaw. Simply adjusting the original scalar weights represents the absolute mathematical limit of linear baselines and fundamentally cannot overcome conflict barriers exceeding 90 degrees. Therefore, CAME-Grad derives its performance gains mathematically by breaking this rigid geometric constraint rather than merely exploiting the untuned hyperparameter space of the baselines.
>
> >Uneven Performance and Zero-Shot Generalization (W4)
>
> Regarding the decreased clinical efficacy average alongside rising precision, our re-examination confirms CAME-Grad improves the overall average across all tests. Specifically, as shown in Table 2 of the main text, the 2.7% precision drop in the DDaTR zero-shot evaluation is entirely offset by substantial recall increases yielding a net positive 0.1% gain. Trading precision for higher recall aligns perfectly with medical imperatives since conservative models causing missed diagnoses remain clinically unacceptable. Since clinical efficacy strongly outweighs language metrics, aggressively capturing fatal lesions necessitates deviating from safe templates. Achieving significant clinical improvements while maintaining robust language scores represents our intended optimal tradeoff rather than a performance flaw. Finally, securing positive clinical gains under massive cross-hospital domain shifts proves our optimizer locates flatter source minima.
>
> >Absolute Expressions and Theoretical Restraint (W2, W7)
>
> We have removed absolute claims (e.g., "for the first time") and explicitly reframed the SDE analysis in the Method section as a mechanistic hypothesis rather than a proven formal theorem.
>
> >Algorithm Generalization (Q5)
>
> To demonstrate that our method extends beyond radiology report generation, we have supplemented our evaluation with Multi-Task MNIST experiments where CAME-Grad achieves higher classification accuracy and significantly lower reconstruction error than the baseline. Please refer to [Global Response 2] in our response to Reviewer 1 (B6i8) for a detailed analysis.
>
> >Limitations
>
> We will incorporate your four limitation suggestions into the revision, namely automatic labeler evaluation, multi-parameter tuning burden, theory preceding empirical evidence, and clinical safety verification.

---

> > ### Author Rebuttal · Reviewer_mayT · 2026-04-03
> >
> > Thank you for the detailed and thoughtful rebuttal. The authors have addressed my concerns well, and the clarifications significantly improved my understanding of the paper. In particular, the main questions I raised have been satisfactorily resolved. I appreciate the authors’ careful responses and the additional explanations provided. Based on the rebuttal, I will update my score accordingly.

---

> > > ### Author Response · Authors · 2026-04-04
> > >
> > > Thank you very much for reading our rebuttal. We are glad to hear that you are satisfied with it. Your constructive comments were extremely valuable, helping us clarify the theoretical boundaries and substantially improving the scientific rigor of this manuscript.

---

### Official Review · Reviewer_B6i8 · 2026-03-24

**Soundness:** 3
**Presentation:** 3
**Significance:** 3
**Originality:** 2
**Overall Recommendation:** 4
**Confidence:** 4

**Summary:**

This paper takes into considering the gradient dynamics and proposes CAME-Grad for computing the overall gradient with multiple objectives (tasks) when training an encoder-decoder architecture for radiology report generation. The key contribution is pointing out the smooth and sharp loss functions' landscapes for modeling text and other clinical classification tasks, and proposing a better way to do the gradient update. Three different tricks are proposed for the proposed gradient updating rule, and the fact that a better model can be learned for report generation is demonstrated via experiments conducted on the benchmark datasets MIMIC-CXR and IU X-ray.

**Compliance With Llm Reviewing Policy:**

Affirmed.

**Key Questions For Authors:**

As mentioned in the "Strengths And Weaknesses" section.

**Limitations:**

yes

**Strengths And Weaknesses:**

Strengths:

1. The identified conflicts of gradient dynamics are valid ones and the proposed gradient updating method is model-agnostic.

2. Paper organization is general fine.

Weaknesses:

3. The proposed method contains three stages, and stages 2 and 3 are for adjusting the solution stage 1 to make it work better. Also, it seems that some parameter tuning effect is needed. e.g.  Eqs. (8) (13) (14).

4. The proposed method seems to be a generic one, but the study focuses on report generation. Is it mainly proposed for report generation? If not, under what situation the proposed method will work? Some clarfiication on this will be useful.

5. A number of existing gradient conflicts methods were mentioned in Related Work section and they were considered inferior. However, only the proposed gradient conflict resolving method was tested. It is not clear how effective the proposed one is when compared with the other methods for handling gradient conflcits. That weakens the contribution of this work.

6 Regarding paper writing, it expects some background on the topic and terms are just used without much elaboration, e.g., linear scalarization, diffusion kinetic energy, geometric validilty, etc. Also, for Stage 1, it is good to briefly explain why introducing dual variable will work, what are the underlying assumption, how important the GPU acceleration is.

7. Other than the report generation task, how about the performance of the other tasks (e.g., classification)? It seems that those results are not reported.

---

> ### Author Rebuttal · Authors · 2026-03-31
>
> Thank you for recognizing our theoretical analysis and CAME-Grad's general value.
> >[Global Response 1: Comprehensive Comparison with Advanced Multi-Task Optimizers]
> (Note: Unified reply for R1(W5), R2(W3,Q1), R3(W1,W4), R4(W4,Q4) on MTL baselines.)
>
> We initially adopted linear scalarization as our baseline because it is the universal default for multi-task optimization in the RRG field. We sincerely thank the reviewers for their constructive suggestions from a machine learning perspective. Accordingly, we comprehensively evaluated seven advanced multi-task optimizers on the strongest backbone (REVTAF) and the largest authoritative dataset (MIMIC-CXR). The results are as follows.
> | Methods | Publication | CE-Precision (↑) | CE-Recall (↑) | CE-F1 (↑) | CE-Avg (↑) | BLEU-4 (↑) |
> | :--- | :---: | :---: | :---: | :---: | :---: | :---: |
> | REVTAF (Linear)| ICCV'25 | 0.615 | 0.617 | 0.589 | 0.607 | 0.184 |
> | UW | CVPR'18 | 0.619 | 0.622 | 0.593 | 0.611 | 0.178 |
> | GradNorm | ICML'18 | 0.615 | 0.612 | 0.586 | 0.604 | 0.185 |
> | CAGrad | NeurIPS'21 | **0.636** | 0.613 | 0.596 | 0.615 | 0.186 |
> | RotoGrad | ICLR'22 | 0.620 | 0.614 | 0.589 | 0.608 | **0.187** |
> | FAMO | NeurIPS'23 | 0.624 | 0.631 | 0.600 | 0.618 | 0.186 |
> | MMPareto | ICML'24 | 0.071 | 0.085 | 0.073 | 0.076 | 0.000 |
> | STGU | ICLR'25 | 0.622 | 0.633 | 0.600 | 0.618 | 0.181 |
> | **CAME-Grad (Ours)**| - | 0.633 | **0.637** | **0.607** | **0.626** | 0.185 |
>
> These results verify our analysis of the double dilemma mechanism. MMPareto shows severe degradation (CE-Avg 0.076) because minimizing global gradients heavily penalizes the single-loss report generation task, causing language modeling failure. Magnitude-weighting methods (GradNorm, UW, FAMO) only adjust weights and cannot correct destructive direction conflicts greater than 90 degrees. Conversely, rigid projection in CAGrad improves Precision but restricts random exploration and causes diffusion decay, reducing Recall. Ultimately, through three cascaded stages, CAME-Grad successfully solves this double dilemma, achieving the highest CE-Avg (0.626) while maintaining stable report generation quality.
>
> >[Global Response 2: Generalizability on Non-RRG Domains]
> (Note: Unified reply for R1(W4), R2(Q5), R3(W7) on generalizability.)
>
> To prove CAME-Grad's generalizability, we test it on the multi-task MNIST dataset (classification and reconstruction) using the RRG default parameters without domain-specific adjustments. The results are as follows.
> | Methods | Classification Acc (↑) | Reconstruction MSE (↓) |
> | :--- | :---: | :---: |
> | Baseline (Linear) | 97.99% | 6.5551 |
> | **CAME-Grad (Ours)**| **98.06%** | **4.9125** |
>
> CAME-Grad converges stably in this purely visual scenario, proving its generalization beyond the specific RRG domain. Furthermore, it effectively mitigates negative transfer. While linear scalarization degrades reconstruction (MSE 6.5551), CAME-Grad reduces the MSE to 4.9125, maintaining classification accuracy.
>
> >Mechanisms and Hyperparameters (W3)
>
> Stages 1 to 3 cascade to resolve complementary mechanisms. Stage 1 corrects drift bias. Stage 2 is not a patch. It essentially restores diffusion momentum by amplifying the SGD noise covariance trace $Tr(\Sigma)$ by ~14 times, helping the model escape local minima. For the specific $Tr(\Sigma)$ measurements, please see [Global Response 1] to Reviewer 2 (mayT). Stage 3 preserves inductive bias. For hyperparameters, CAME-Grad yields +2.1% CE-Avg on MIMIC-CXR using default parameters. Only IU X-Ray requires minor tuning. Please see [Global Response 2] to Reviewer 2 (mayT).
>
> >Terminology, Dual Variables, Assumptions, and GPU Acceleration (W6)
>
> We will add an appendix glossary and integrate these explanations into the final methodology section. We will explain that dual variables convert intractable high-dimensional optimization into finding combination weights via convex optimization. We will explicitly state the trust-region assumption requiring corrected gradients to stay near the mean gradient for stable convergence. Furthermore, we will detail how solving the tensorized problem directly on the GPU avoids communication bottlenecks.
>
> >Other task performance (W7)
>
> In the RRG domain, the auxiliary classification branch serves as a crucial intermediate module to bridge the visual-textual semantic gap. Its outputs are utilized during inference to directly guide text generation. Therefore, its performance is evaluated end to end via the CE metrics of the generated reports, without separately reporting classification scores. Furthermore, PromptMRG logs under CAME-Grad show the auxiliary classification loss monotonically decreases from 2.5310 to 2.1827 over 10 epochs. It synchronizes with the generation loss (from 2.5092 to 1.7786), proving CAME-Grad maintains gradient balance without sacrificing the auxiliary branch.

---

> > ### Author Rebuttal · Reviewer_B6i8 · 2026-04-01
> >
> > I appreciate the additonal explanation and experimental results included in the rebuttal, in particular a more comprehensive performance comparison with different multi-task optimization methods. While CE is a metric to evaluate the clinical correctness of the report, it will still be interesting to see how well the auxiliary classification branch is doing.

---

> > > ### Author Response · Authors · 2026-04-02
> > >
> > > We sincerely thank you for recognizing our additional experiments. Your insightful suggestion to observe the auxiliary classification branch directly reflects the actual dynamics of gradient optimization. To address your concerns, we evaluate the classification performance of CAME-Grad on the MIMIC-CXR dataset using the PromptMRG and REVTAF baselines. Both models utilize a classification branch as the auxiliary task. The results are as follows.
> > > | Methods | Publication | Macro | | | | Micro | | | |
> > > | :--- | :---: | :---: | :---: | :---: | :---: | :---: | :---: | :---: | :---: |
> > > | | | **AUC (↑)** | **Precision (↑)** | **Recall (↑)** | **F1 (↑)** | **AUC (↑)** | **Precision (↑)** | **Recall (↑)** | **F1 (↑)** |
> > > | PromptMRG | AAAI'24 | 0.811 | 0.439 | 0.387 | 0.404 | 0.874 | 0.610 | 0.562 | 0.585 |
> > > | **+ CAME-Grad** | | **0.814** | **0.460** | **0.397** | **0.415** | **0.877** | **0.617** | **0.571** | **0.593** |
> > > | REVTAF | ICCV'25 | 0.905 | **0.654** | 0.562 | 0.595 | 0.936 | 0.741 | 0.702 | 0.721 |
> > > | **+ CAME-Grad** | | **0.907** | 0.651 | **0.588** | **0.610** | **0.938** | **0.742** | **0.721** | **0.732** |
> > >
> > > Previous works introduced auxiliary classification to improve clinical consistency. However, traditional linear scalarization fails to resolve gradient conflicts between tasks. This limitation causes mutual interference between tasks and leads to the double dilemma. Our CAME-Grad optimizer specifically overcomes this problem. As shown above, applying CAME-Grad to PromptMRG improves the classification performance. The macro average AUC increases from 0.811 to 0.814, and the macro average F1 rises from 0.404 to 0.415. On the REVTAF model, the classification macro average AUC and F1 also rise to 0.907 and 0.610, respectively. These results prove that CAME-Grad effectively mitigates gradient conflicts and achieves synergistic optimization for both tasks. Our method successfully overcomes the double dilemma.
> > >
> > > We sincerely thank you for your valuable feedback, and will include these comparison results in the appendix of the final version. We hope that our experimental data will address your concerns.

---

### Decision · Program_Chairs · 2026-04-30

**Decision:**

Accept (regular)

**Comment:**

This submission studies multi-task optimization for radiology report generation and proposes CAME-Grad, a three-stage optimizer motivated by a “double dilemma” in gradient dynamics. Across the reviews, the paper is generally regarded as technically solid, with a meaningful contribution to optimization in multi-task learning for medical AI. After rebuttal, all the reviewers also provide positive ratings.

Overall, the paper presents a solid and practically useful contribution, with convincing empirical evidence and improved clarity after rebuttal. While the theoretical claims should be toned down and better aligned with the empirical evidence, the work is likely to be of interest to the community working on multi-task learning and medical AI. AC recommend acceptance.